# Multi-Label Learning with Stronger Consistency Guarantees

**Anqi Mao**
Courant Institute
New York, NY 10012
aqmao@cims.nyu.edu

**Mehryar Mohri**
Google Research & CIMS
New York, NY 10011
mohri@google.com

**Yutao Zhong**
Courant Institute
New York, NY 10012
yutao@cims.nyu.edu

## Abstract

We present a detailed study of surrogate losses and algorithms for multi-label learning, supported by $\mathcal{H}$-consistency bounds. We first show that, for the simplest form of multi-label loss (the popular Hamming loss), the well-known consistent binary relevance surrogate suffers from a sub-optimal dependency on the number of labels in terms of $\mathcal{H}$-consistency bounds, when using smooth losses such as logistic losses. Furthermore, this loss function fails to account for label correlations. To address these drawbacks, we introduce a novel surrogate loss, *multi-label logistic loss*, that accounts for label correlations and benefits from label-independent $\mathcal{H}$-consistency bounds. We then broaden our analysis to cover a more extensive family of multi-label losses, including all common ones and a new extension defined based on linear-fractional functions with respect to the confusion matrix. We also extend our multi-label logistic losses to more comprehensive multi-label comp-sum losses, adapting comp-sum losses from standard classification to the multi-label learning. We prove that this family of surrogate losses benefits from $\mathcal{H}$-consistency bounds, and thus Bayes-consistency, across any general multi-label loss. Our work thus proposes a unified surrogate loss framework benefiting from strong consistency guarantees for any multi-label loss, significantly expanding upon previous work which only established Bayes-consistency and for specific loss functions. Additionally, we adapt constrained losses from standard classification to multi-label constrained losses in a similar way, which also benefit from $\mathcal{H}$-consistency bounds and thus Bayes-consistency for any multi-label loss. We further describe efficient gradient computation algorithms for minimizing the multi-label logistic loss.

## 1 Introduction

Supervised learning methods often assign a single label to each instance. However, real-world data exhibits a more complex structure, with objects belonging to multiple categories simultaneously. Consider a video about sports training, which could be categorized as both 'health' and 'athletics,' or a culinary blog post tagged with 'cooking' and 'nutrition'. As a result, multi-label learning [McCallum, 1999, Schapire and Singer, 2000] has become increasingly important, leading to the development of various interesting and effective approaches, predominantly experimental in nature, in recent years [Elisseeff and Weston, 2001, Deng et al., 2011, Petterson and Caetano, 2011, Kapoor et al., 2012].

Although there is a rich literature on multi-label learning (see [Zhang and Zhou, 2013] and [Bogatinovski et al., 2022] for detailed surveys), only a few studies focus on the theoretical analysis of multi-label learning, particularly the study of the Bayes-consistency of surrogate losses [Zhang, 2004a,b, Bartlett et al., 2006, Tewari and Bartlett, 2007, Steinwart, 2007].

38th Conference on Neural Information Processing Systems (NeurIPS 2024).

Gao and Zhou [2011] initiated the study of Bayes-consistency in multi-label learning with respect to Hamming loss and (partial) ranking loss. They provided negative results for ranking loss, demonstrating that no convex and differentiable pairwise surrogate loss is Bayes-consistent for that multi-label loss. They also showed that the *binary relevance* method, which learns an independent binary classifier for each of the $l$ labels, is Bayes-consistent with respect to the Hamming loss. Dembczynski et al. [2011] further demonstrated that under the assumption of conditionally independent labels, the *binary relevance* method is also Bayes-consistent with respect to the $F_\beta$ measure loss. However, they noted that it can perform arbitrarily poorly when this assumption does not hold. Dembczynski et al. [2012] provided a positive result for the (partial) ranking loss by showing that the simpler univariate variants of smooth surrogate losses are Bayes-consistent with respect to it. Additionally, Zhang et al. [2020] proposed a family of Bayes-consistent surrogate losses for the $F_\beta$ measure by reducing the $F_\beta$ learning problem to a set of binary class probability estimation problems. This approach was motivated by the consistent output coding scheme in [Ramaswamy et al., 2014] for general multiclass problems. Other works have studied generalization bounds in multi-label learning [Yu et al., 2014, Wydmuch et al., 2018, Wu and Zhu, 2020, Wu et al., 2021, 2023, Busa-Fekete et al., 2022].

Another related topic is the characterization of the Bayes classifier and corresponding Bayes-consistent plug-in algorithm in multi-label learning. This includes the characterization of the Bayes classifier for subset $0/1$ loss and Hamming loss in [Cheng et al., 2010] and the characterization of the Bayes classifier for $F_1$ measure in [Dembczynski et al., 2011]. Dembczynski et al. [2013], Waegeman et al. [2014] further extended the results in [Dembczynski et al., 2011] by designing a Bayes-consistent plug-in algorithm for the $F_\beta$ measure. Koyejo et al. [2015] characterized the Bayes classifier for general linear fractional losses with respect to the confusion matrix and designed the corresponding plug-in algorithms in the empirical utility maximization (EUM) framework. In this framework, the measures are directly defined as functions of the population, in contrast to a loss function that is defined as a function over a single instance in the decision theoretic analysis (DTA) framework [Ye et al., 2012]. Menon et al. [2019] studied the Bayes-consistency of various reduction methods with respect to Precision@$\kappa$ and Recall@$\kappa$ in multi-label learning. However, all these publications only established Bayes-consistency for specific loss functions. Can we derive a unified surrogate loss framework that is Bayes-consistent for any multi-label loss?

Furthermore, as Awasthi, Mao, Mohri, and Zhong [2022a,b] pointed out, Bayes-consistency is an asymptotic guarantee and does not provide convergence guarantees. It also applies only to the family of all measurable functions unlike the restricted hypothesis sets typically used in practice. Instead, they proposed a stronger guarantee known as $\mathcal{H}$-*consistency bounds*, which are both non-asymptotic and account for the hypothesis set while implying Bayes-consistency. These guarantees provide upper bounds on the target estimation error in terms of the surrogate estimation error. Can we leverage this state-of-the-art consistency guarantee when designing surrogate loss functions for multi-label learning?

Moreover, one of the main concerns in multi-label learning is label correlations (see [Dembczyński et al., 2012]). For the simplest form of multi-label loss, the popular Hamming loss, the existing Bayes-consistent binary relevance surrogate fails to account for label correlations. Can we design consistent loss functions that effectively account for label correlations as well?

**Our Contributions.** This paper directly addresses these key questions in multi-label learning. We present a detailed study of surrogate losses and algorithms for multi-label learning, supported by $\mathcal{H}$-consistency bounds.

In Section 3, we first show that for the simplest form of multi-label loss, the popular Hamming loss, the well-known consistent binary relevance surrogate, when using smooth losses such as logistic losses, suffers from a sub-optimal dependency on the number of labels in terms of $\mathcal{H}$-consistency bounds. Furthermore, this loss function fails to account for label correlations.

To address these drawbacks, we introduce a novel surrogate loss, *multi-label logistic loss*, that accounts for label correlations and benefits from label-independent $\mathcal{H}$-consistency bounds (Section 4). We then broaden our analysis to cover a more extensive family of multi-label losses, including all common ones and a new extension defined based on linear-fractional functions with respect to the confusion matrix (Section 5).

In Section 6, we also extend our multi-label logistic losses to more comprehensive multi-label comp-sum losses, adapting comp-sum losses from standard classification to the multi-label learning.

We prove that this family of surrogate losses benefits from $\mathcal{H}$-consistency bounds, and thus Bayes-consistency, across any general multi-label loss. Our work thus proposes a unified surrogate loss framework that is Bayes-consistent for any multi-label loss, significantly expanding upon previous work which only established consistency for specific loss functions.

Additionally, we adapt constrained losses from standard classification to multi-label constrained losses in a similar way, which also benefit from $\mathcal{H}$-consistency bounds and thus Bayes-consistency for any multi-label loss (Section 7). We further describe efficient gradient computation algorithms for minimizing the multi-label logistic loss (Section 8).

## 2    Preliminaries

**Multi-label learning.** We consider the standard multi-label learning setting. Let $\mathcal{X}$ be the input space and $\mathcal{Y} = \{+1, -1\}^l$ the set of all possible labels or tags, where $l$ is a finite number. For example, $\mathcal{X}$ can be a set of images, and $\mathcal{Y}$ can be a set of $l$ pre-given tags (such as 'flowers', 'shoes', or 'books') that can be associated with each image in the image tagging problem. Let $n = |\mathcal{Y}|$. For any instance $x \in \mathcal{X}$ and its associated label $y = (y_1, \ldots, y_l) \in \mathcal{Y}$, if $y_i = +1$, we say that label $i$ is relevant to $x$. Otherwise, it is not relevant. Let $[l] = \{1, \ldots, l\}$. Given a sample $S$ drawn i.i.d. according to some distribution $\mathcal{D}$ over $\mathcal{X} \times \mathcal{Y}$, the goal of multi-label learning is to learn a hypothesis $h \colon \mathcal{X} \times [l] \to \mathbb{R}$ to minimize the generalization error defined by a multi-label loss function $\mathsf{L} \colon \mathcal{H}_{\mathrm{all}} \times \mathcal{X} \times \mathcal{Y} \to \mathbb{R}$,

$$\mathcal{R}_{\mathsf{L}}(h) = \mathbb{E}_{(x,y) \sim \mathcal{D}} \left[ \mathsf{L}(h, x, y) \right], \tag{1}$$

where $\mathcal{H}_{\mathrm{all}}$ is the family of all measurable hypotheses. For convenience, we abusively denote the scoring vector by $h(x) = (h(x, 1), \ldots, h(x, l))$. Given a hypothesis set $\mathcal{H} \subset \mathcal{H}_{\mathrm{all}}$, we denote by $\mathcal{R}_{\mathsf{L}}^*(\mathcal{H}) = \inf_{h \in \mathcal{H}} \mathcal{R}(h)$ the best-in-class error. We refer to the difference $\mathcal{R}_{\mathsf{L}}(h) - \mathcal{R}_{\mathsf{L}}^*(\mathcal{H})$ as the *estimation error*, which is termed the *excess error* when $\mathcal{H} = \mathcal{H}_{\mathrm{all}}$. Let $\mathrm{sign} \colon t \mapsto 1_{t \geq 0} - 1_{t < 0}$ be the sign function, and let $t \colon \mathcal{X} \to \mathbb{R}_+$ be a threshold function. The target loss function $\mathsf{L}$ can be typically given by a function $\overline{\mathsf{L}}$ mapping from $\mathcal{Y} \times \mathcal{Y}$ to real numbers:

$$\mathsf{L}(h, x, y) = \overline{\mathsf{L}}(\mathsf{h}(x), y), \tag{2}$$

where $\mathsf{h}(x) \coloneqq [\mathsf{h}_1(x), \ldots, \mathsf{h}_l(x)] \in \mathcal{Y}$ is the prediction for the input $x \in \mathcal{X}$ and $\mathsf{h}_i(x) = \mathrm{sign}(h(x, i))$ for any $i \in [l]$. There are many multi-label loss functions, such as Hamming loss, (partial) ranking loss, $F_1$ and the more general $F_\beta$ measure loss, subset $0/1$ loss, precision@$\kappa$, recall@$\kappa$, etc. [Zhang and Zhou, 2013]. Among these, several loss functions are defined based on the prediction of the hypothesis $\mathsf{h}(x)$, while others are based on the scoring vector $h(x)$. We will specifically consider the first type of multi-label loss in the form given in (2), which is based on some 'distance' between the prediction and the true label. This includes all the loss functions previously mentioned (see Section 5 for a list of several common multi-label losses in this family) but excludes the (partial) ranking loss, which is defined based on pairwise scores. For convenience, we may alternatively refer to $\overline{\mathsf{L}}$ or its induced $\mathsf{L}$ as the multi-label loss. Without loss of generality, we assume that $\overline{\mathsf{L}} \in [0, 1]$, which can be achieved through normalization. We also denote by $\mathsf{L}_{\max} = \max_{y', y} \overline{\mathsf{L}}(y', y)$. Our analysis is general and adapts to any multi-label loss $\overline{\mathsf{L}}$. Note that we adhere to the decision-theoretic analysis (DTA) framework, in which a loss function is defined over a single instance, and the measure is the expected loss, also known as the generalization error (the expectation of the loss function over samples). Another popular framework is empirical utility maximization (EUM), where measures are defined directly as functions of the population, that is, as a function of the expectation over samples.

**Surrogate risk minimization and consistency.** Minimizing the multi-label loss $\mathsf{L}$ directly is computationally hard for most hypothesis sets because it is discrete and non-convex. A common method involves minimizing a smooth surrogate loss function $\widetilde{\mathsf{L}} \colon \mathcal{H}_{\mathrm{all}} \times \mathcal{X} \times \mathcal{Y} \to \mathbb{R}$, which is the main focus of this paper. Minimizing a surrogate loss directly leads to an algorithm for multi-label learning. A desirable guarantee for the surrogate loss in multi-label learning is *Bayes-consistency* [Zhang, 2004a,b, Bartlett et al., 2006, Tewari and Bartlett, 2007, Steinwart, 2007, Gao and Zhou, 2011]. That is, minimizing the surrogate loss over the family of all measurable functions leads to the minimization of the multi-label loss over the same family:

**Definition 2.1.** A surrogate loss $\widetilde{\mathsf{L}}$ is said to be *Bayes-consistent* with respect to a multi-label loss $\mathsf{L}$ if the following holds for any distribution and all given sequences of hypotheses $\{h_n\}_{n \in \mathbb{N}} \subset \mathcal{H}_{\mathrm{all}}$:

$$\left( \lim_{n \to +\infty} \mathcal{R}_{\widetilde{\mathsf{L}}}(h_n) - \mathcal{R}_{\widetilde{\mathsf{L}}}^*(\mathcal{H}_{\mathrm{all}}) = 0 \right) \implies \left( \lim_{n \to +\infty} \mathcal{R}_{\mathsf{L}}(h_n) - \mathcal{R}_{\mathsf{L}}^*(\mathcal{H}_{\mathrm{all}}) = 0 \right). \tag{3}$$

As pointed out by Awasthi, Mao, Mohri, and Zhong [2022a,b] (see also [Long and Servedio, 2013, Zhang and Agarwal, 2020, Awasthi et al., 2021a,b, Mao et al., 2023d,c,a,b,e, Awasthi et al., 2023, 2024, Mao et al., 2024a,b,c,h,g,e,h,d,f, Mohri et al., 2024, Cortes et al., 2024]), Bayes-consistency is an asymptotic guarantee that cannot provide any guarantee for approximate minimizers; it also applies only to the family of all measurable functions and does not consider the hypothesis sets typically used in practice. Instead, they propose a stronger guarantee known as $\mathcal{H}$-*consistency bounds*, which are both non-asymptotic and dependent on the hypothesis set, and imply Bayes-consistency when $\mathcal{H} = \mathcal{H}_{\mathrm{all}}$. These guarantees provide upper bounds on the target estimation error in terms of the surrogate estimation error. In the multi-label learning scenario, they can be formulated as follows:

**Definition 2.2.** A surrogate loss $\widetilde{\mathsf{L}}$ is said to admit an $\mathcal{H}$-*consistency bound* with respect to a multi-label loss $\mathsf{L}$ if the following condition holds for any distribution and for all hypotheses $h \in \mathcal{H}$, given a concave function $\Gamma \colon \mathbb{R}_+ \to \mathbb{R}_+$ with $\Gamma(0) = 0$:

$$\mathcal{R}_{\mathsf{L}}(h) - \mathcal{R}_{\mathsf{L}}^*(\mathcal{H}) + \mathcal{M}_{\mathsf{L}}(\mathcal{H}) \le \Gamma\big(\mathcal{R}_{\widetilde{\mathsf{L}}}(h) - \mathcal{R}_{\widetilde{\mathsf{L}}}^*(\mathcal{H}) + \mathcal{M}_{\widetilde{\mathsf{L}}}(\mathcal{H})\big). \tag{4}$$

The quantities $\mathcal{M}_{\widetilde{\mathsf{L}}}(\mathcal{H})$ appearing in the bounds are called minimizability gaps, which measure the difference between the best-in-class error and the expected best pointwise error for a loss function $\widetilde{\mathsf{L}}$ and a hypothesis set $\mathcal{H}$:

$$\mathcal{M}_{\widetilde{\mathsf{L}}}(\mathcal{H}) = \mathcal{R}_{\widetilde{\mathsf{L}}}^*(\mathcal{H}) - \mathbb{E}_x\left[\inf_{h \in \mathcal{H}}\left(\mathbb{E}_{y|x}\big[\widetilde{\mathsf{L}}(h, x, y)\big]\right)\right] \ge 0. \tag{5}$$

These are inherent quantities depending on the distribution and hypothesis set, which we cannot hope to minimize. Since $\Gamma$ is concave and $\Gamma(0) = 0$, $\Gamma$ is sub-additive and an $\mathcal{H}$-consistency bound (4) implies that: $\mathcal{R}_{\mathsf{L}}(h) - \mathcal{R}_{\mathsf{L}}^*(\mathcal{H}) + \mathcal{M}_{\mathsf{L}}(\mathcal{H}) \le \Gamma\big(\mathcal{R}_{\widetilde{\mathsf{L}}}(h) - \mathcal{R}_{\widetilde{\mathsf{L}}}^*(\mathcal{H})\big) + \Gamma\big(\mathcal{M}_{\widetilde{\mathsf{L}}}(\mathcal{H})\big)$. Therefore, when the surrogate estimation error $\big(\mathcal{R}_{\widetilde{\mathsf{L}}}(h) - \mathcal{R}_{\widetilde{\mathsf{L}}}^*(\mathcal{H})\big)$ is minimized to $\epsilon$, the target estimation error $(\mathcal{R}_{\mathsf{L}}(h) - \mathcal{R}_{\mathsf{L}}^*(\mathcal{H}))$ is upper bounded by $\Gamma(\epsilon) + \Gamma\big(\mathcal{M}_{\widetilde{\mathsf{L}}}(\mathcal{H})\big)$. The minimizability gaps vanish when $\mathcal{H} = \mathcal{H}_{\mathrm{all}}$ or in more general realizable cases, such as when $\mathcal{R}_{\widetilde{\mathsf{L}}}^*(\mathcal{H}) = \mathcal{R}_{\widetilde{\mathsf{L}}}^*(\mathcal{H}_{\mathrm{all}})$ [Steinwart, 2007, Awasthi, Mao, Mohri, and Zhong, 2022b, Mao, Mohri, and Zhong, 2023f]. In these cases, $\mathcal{H}$-consistency bounds imply the $\mathcal{H}$-consistency of a surrogate loss $\widetilde{\mathsf{L}}$ with respect to a multi-label loss $\mathsf{L}$: $\mathcal{R}_{\widetilde{\mathsf{L}}}(h) - \mathcal{R}_{\widetilde{\mathsf{L}}}^*(\mathcal{H}) \le \epsilon \implies \mathcal{R}_{\mathsf{L}}(h) - \mathcal{R}_{\mathsf{L}}^*(\mathcal{H}) \le \Gamma(\epsilon)$, for any $\epsilon \ge 0$. The minimizability gap $\mathcal{M}_{\widetilde{\mathsf{L}}}(\mathcal{H})$ is upper bounded by the approximate error $\mathcal{A}_{\widetilde{\mathsf{L}}}(\mathcal{H}) = \mathcal{R}_{\widetilde{\mathsf{L}}}^*(\mathcal{H}) - \mathbb{E}_x\big[\inf_{h \in \mathcal{H}_{\mathrm{all}}}\big(\mathbb{E}_{y|x}\big[\widetilde{\mathsf{L}}(h, x, , y)\big]\big)\big]$ and is generally a finer quantity [Mao et al., 2023f]. Thus, $\mathcal{H}$-consistency bounds are more informative, more favorable, and stronger than excess error bounds, and they imply these bounds when $\mathcal{H} = \mathcal{H}_{\mathrm{all}}$.

Next, we will study surrogate loss functions and algorithms for multi-label learning, supported by $\mathcal{H}$-consistency bounds, the state-of-the-art consistency guarantee for surrogate risk minimization.

## 3 Existing consistent surrogates for the Hamming loss

In the section, we consider the simplest form of multi-label loss, the Hamming loss, defined as:

$$\forall (h, x, y) \in \mathcal{H} \times \mathcal{X} \times \mathcal{Y}, \quad \mathsf{L}_{\mathrm{ham}}(h, x, y) = \overline{\mathsf{L}}_{\mathrm{ham}}(\mathsf{h}(x), y), \text{ where } \overline{\mathsf{L}}_{\mathrm{ham}}(y', y) = \sum_{i=1}^{l} 1_{y_i \ne y_i'}. \tag{6}$$

The existing Bayes-consistent surrogate loss function is to transform the multi-label learning into $l$ independent binary classification tasks [Gao and Zhou, 2011], defined as for all $(h, x, y) \in \mathcal{H} \times \mathcal{X} \times \mathcal{Y}$,

$$\widetilde{\mathsf{L}}_{\mathrm{br}}(h, x, y) = \sum_{i=1}^{l} \Phi(y_i h(x, i)), \tag{7}$$

where $\Phi \colon \mathbb{R} \to \mathbb{R}_+$ is a binary margin-based loss function, such as the logistic loss $u \mapsto \log(1 + e^{-u})$. The algorithm that minimizes this surrogate loss is known as *binary relevance* [Zhang and Zhou, 2013], which learns an independent binary classifier for each of the $l$ labels. Gao and Zhou [2011, Theorem 15] shows that $\widetilde{\mathsf{L}}_{\mathrm{br}}$ is Bayes-consistent with respect to $\mathsf{L}_{\mathrm{ham}}$ if $\Phi$ is Bayes-consistent with respect to $\ell_{0-1} \colon (f, x, y) \mapsto 1_{y \ne \mathrm{sign}(f(x))}$, the binary zero-one loss. Here, we prove a stronger result that $\widetilde{\mathsf{L}}_{\mathrm{br}}$ admits an $\mathcal{H}$-consistency bound with respect to $\mathsf{L}_{\mathrm{ham}}$ with a functional form $l\Gamma\big(\frac{\cdot}{l}\big)$ if $\Phi$ admits an $\mathcal{H}$-consistency bounds with respect to $\ell_{0-1}$ with a functional form $\Gamma(\cdot)$. Let $\mathcal{F}$ be a hypothesis set consisting of functions mapping from $\mathcal{X}$ to $\mathbb{R}$.

**Theorem 3.1.** *Let $\mathcal{H} = \mathcal{F}^l$. Assume that the following $\mathcal{F}$-consistency bound holds in the binary classification, for some concave function $\Gamma: \mathbb{R} \to \mathbb{R}_+$:*

$$\forall f \in \mathcal{F}, \quad \mathcal{R}_{\ell_{0-1}}(f) - \mathcal{R}^*_{\ell_{0-1}}(\mathcal{F}) + \mathcal{M}_{\ell_{0-1}}(\mathcal{F}) \leq \Gamma(\mathcal{R}_{\Phi}(f) - \mathcal{R}^*_{\Phi}(\mathcal{F}) + \mathcal{M}_{\Phi}(\mathcal{F})). \quad (8)$$

*Then, the following $\mathcal{H}$-consistency bound holds in the multi-label learning: for all $h \in \mathcal{H}$,*

$$\mathcal{R}_{\mathsf{L}_{\mathrm{ham}}}(h) - \mathcal{R}^*_{\mathsf{L}_{\mathrm{ham}}}(\mathcal{H}) + \mathcal{M}_{\mathsf{L}_{\mathrm{ham}}}(\mathcal{H}) \leq l\Gamma\!\left( \frac{\mathcal{R}_{\widetilde{\mathsf{L}}_{\mathrm{br}}}(h) - \mathcal{R}^*_{\widetilde{\mathsf{L}}_{\mathrm{br}}}(\mathcal{H}) + \mathcal{M}_{\widetilde{\mathsf{L}}_{\mathrm{br}}}(\mathcal{H})}{l} \right). \quad (9)$$

The proof is included in Appendix A. We say that a hypothesis set $\mathcal{F}$ is *complete* if $\{f(x): f \in \mathcal{F}\} = \mathbb{R}$ for all $x \in \mathcal{X}$. This notion of completeness is broadly applicable and holds for commonly used hypothesis sets in practice, including linear hypotheses, multi-layer feed-forward neural networks, and all measurable functions. For such complete hypothesis sets $\mathcal{F}$ and with smooth functions $\Phi$ like the logistic loss function, $\Gamma$ admits a square root dependency in the binary classification [Awasthi et al., 2022a, Mao et al., 2024h]. Thus, by Theorem 3.1, we obtain the following result.

**Corollary 3.2.** *Let $\mathcal{H} = \mathcal{F}^l$. Assume that $\mathcal{F}$ is complete and $\Phi(u) = \log(1 + e^{-u})$. Then, the following $\mathcal{H}$-consistency bound holds in the multi-label learning: for all $h \in \mathcal{H}$,*

$$\mathcal{R}_{\mathsf{L}_{\mathrm{ham}}}(h) - \mathcal{R}^*_{\mathsf{L}_{\mathrm{ham}}}(\mathcal{H}) + \mathcal{M}_{\mathsf{L}_{\mathrm{ham}}}(\mathcal{H}) \leq l^{\frac{1}{2}}\!\left( \mathcal{R}_{\widetilde{\mathsf{L}}_{\mathrm{br}}}(h) - \mathcal{R}^*_{\widetilde{\mathsf{L}}_{\mathrm{br}}}(\mathcal{H}) + \mathcal{M}_{\widetilde{\mathsf{L}}_{\mathrm{br}}}(\mathcal{H}) \right)^{\frac{1}{2}}. \quad (10)$$

Since $t \mapsto t^{\frac{1}{2}}$ is sub-additive, the right-hand side of the $\mathcal{H}$-consistency bound in Corollary 3.2 can be further upper bounded by $l^{\frac{1}{2}}\!\left( \mathcal{R}_{\widetilde{\mathsf{L}}_{\mathrm{br}}}(h) - \mathcal{R}^*_{\widetilde{\mathsf{L}}_{\mathrm{br}}}(\mathcal{H}) \right)^{\frac{1}{2}} + l^{\frac{1}{2}}\!\left( \mathcal{M}_{\widetilde{\mathsf{L}}_{\mathrm{br}}}(\mathcal{H}) \right)^{\frac{1}{2}}$. This implies that when the estimation error of the surrogate loss $\widetilde{\mathsf{L}}_{\mathrm{br}}$ is reduced to $\epsilon$, the corresponding estimation error of the Hamming loss is upper bounded by $l^{\frac{1}{2}}\epsilon^{\frac{1}{2}} + l^{\frac{1}{2}}\!\left( \mathcal{M}_{\widetilde{\mathsf{L}}_{\mathrm{br}}}(\mathcal{H}) \right)^{\frac{1}{2}} - \mathcal{M}_{\mathsf{L}_{\mathrm{ham}}}(\mathcal{H})$. In the nearly realizable cases where minimizability gaps are negligible, this upper bound approximates to

$$\mathcal{R}_{\mathsf{L}_{\mathrm{ham}}}(h) - \mathcal{R}^*_{\mathsf{L}_{\mathrm{ham}}}(\mathcal{H}) \leq l^{\frac{1}{2}}\epsilon^{\frac{1}{2}}. \quad (11)$$

Therefore, as the number of labels $l$ increases, the bound becomes less favorable. Furthermore, the loss function $\widetilde{\mathsf{L}}_{\mathrm{br}}$ clearly fails to account for the inherent correlations among labels. For instance, 'coffee' and 'mug' are more likely to co-occur than 'coffee' and 'umbrella'. Additionally, $\widetilde{\mathsf{L}}_{\mathrm{br}}$ is only Bayes-consistent with respect to the Hamming loss and cannot yield risk-minimizing predictions for other multi-label losses such as subset $0/1$ loss or $F_\beta$-measure loss [Dembczyński et al., 2012]. To address these drawbacks, we will introduce a new surrogate loss in the next section.

## 4    Multi-label logistic loss

In this section, we define a new surrogate loss for Hamming loss in multi-label learning that accounts for label correlations and benefits from label-independent $\mathcal{H}$-consistency bounds. This loss function can be viewed as a generalization of the (multinomial) logistic loss [Verhulst, 1838, 1845, Berkson, 1944, 1951], used in standard classification, to multi-label learning. Thus, we will refer to it as *multi-label logistic loss*. It is defined as follows: for all $(h, x, y) \in \mathcal{H} \times \mathcal{X} \times \mathcal{Y}$,

$$\widetilde{\mathsf{L}}_{\log}(h, x, y) = \sum_{y' \in \mathcal{Y}} \left( 1 - \overline{\mathsf{L}}_{\mathrm{ham}}(y', y) \right) \log\!\left( \sum_{y'' \in \mathcal{Y}} e^{\sum_{i=1}^{l}(y''_i - y'_i)h(x,i)} \right). \quad (12)$$

This formulation can be interpreted as a weighted logistic loss, where $\left( 1 - \overline{\mathsf{L}}_{\mathrm{ham}}(\cdot, y) \right)$ serves as a weight vector. Additionally, the term $\sum_{i=1}^{l}(y''_i - y'_i)h(x, i)$ represents the difference in the scores between the label $y'$ and any other label $y''$, where these scores account for the correlations among the labels $y_i$ within the logarithmic function.

The next result shows that the multi-label logistic loss benefits from a favorable $\mathcal{H}$-consistency bound with respect to $\mathsf{L}_{\mathrm{ham}}$, without dependency on the number of labels $l$. We assume that $\mathcal{H} = \mathcal{F}^l$ and $\mathcal{F}$ is complete, conditions that typically hold in practice.

**Theorem 4.1.** *Let $\mathcal{H} = \mathcal{F}^l$. Assume that $\mathcal{F}$ is complete. Then, the following $\mathcal{H}$-consistency bound holds in the multi-label learning: for all $h \in \mathcal{H}$,*

$$\mathcal{R}_{\mathsf{L}_{\mathrm{ham}}}(h) - \mathcal{R}^*_{\mathsf{L}_{\mathrm{ham}}}(\mathcal{H}) + \mathcal{M}_{\mathsf{L}_{\mathrm{ham}}}(\mathcal{H}) \leq 2\!\left( \mathcal{R}_{\widetilde{\mathsf{L}}_{\log}}(h) - \mathcal{R}^*_{\widetilde{\mathsf{L}}_{\log}}(\mathcal{H}) + \mathcal{M}_{\widetilde{\mathsf{L}}_{\log}}(\mathcal{H}) \right)^{\frac{1}{2}}. \quad (13)$$

Since $t \mapsto t^{\frac{1}{2}}$ is sub-additive, the right-hand side of the $\mathcal{H}$-consistency bound in Theorem 4.1 can be further upper bounded by $2\left(\mathcal{R}_{\widetilde{\mathsf{L}}_{\log}}(h) - \mathcal{R}^*_{\widetilde{\mathsf{L}}_{\log}}(\mathcal{H})\right)^{\frac{1}{2}} + 2\left(\mathcal{M}_{\widetilde{\mathsf{L}}_{\log}}(\mathcal{H})\right)^{\frac{1}{2}}$. This implies that when the estimation error of the surrogate loss $\widetilde{\mathsf{L}}_{\log}$ is reduced up to $\epsilon$, the corresponding estimation error of the Hamming loss is upper bounded by $2\epsilon^{\frac{1}{2}} + 2\left(\mathcal{M}_{\widetilde{\mathsf{L}}_{\log}}(\mathcal{H})\right)^{\frac{1}{2}} - \mathcal{M}_{\mathsf{L}_{\mathrm{ham}}}(\mathcal{H})$. In the nearly realizable cases where minimizability gaps are negligible, this upper bound approximates to

$$\mathcal{R}_{\mathsf{L}_{\mathrm{ham}}}(h) - \mathcal{R}^*_{\mathsf{L}_{\mathrm{ham}}}(\mathcal{H}) \leq 2\epsilon^{\frac{1}{2}}. \tag{14}$$

Therefore, the bound is independent of the number of labels $l$. This contrasts with the bound for $\widetilde{\mathsf{L}}_{\mathrm{br}}$ shown in (11), where a label-dependent factor $\ell^{\frac{1}{2}}$ replaces the constant factor 2, making it significantly less favorable.

The proof of Theorem 4.1 is included in Appendix B.2. We first present a general tool (Theorem B.1) in Appendix B.1, which shows that to derive $\mathcal{H}$-consistency bounds in multi-label learning with a concave function $\Gamma$, it is only necessary to upper bound the conditional regret of the target multi-label loss by that of the surrogate loss with the same $\Gamma$. This generalizes [Awasthi, Mao, Mohri, and Zhong, 2022b, Theorem 2] in standard multi-class classification to multi-label learning. Next, we characterize the conditional regret of the target multi-label loss, such as Hamming loss, in Lemma B.2 found in Appendix B.1, under the given assumption. By using Lemma B.2, we upper bound the conditional regret of $\mathsf{L}_{\mathrm{ham}}$ by that of the surrogate loss $\widetilde{\mathsf{L}}_{\log}$ with a concave function $\Gamma(t) = 2\sqrt{t}$.

When $\mathcal{H} = \mathcal{H}_{\mathrm{all}}$, minimizability gaps $\mathcal{M}_{\widetilde{\mathsf{L}}_{\log}}(\mathcal{H})$ and $\mathcal{M}_{\mathsf{L}_{\mathrm{ham}}}(\mathcal{H})$ vanish, Theorem 4.1 implies excess error bound and Bayes-consistency of multi-label logistic loss with respect to the Hamming loss.

**Corollary 4.2.** *The following excess error bound holds in the multi-label learning: for all $h \in \mathcal{H}_{\mathrm{all}}$,*

$$\mathcal{R}_{\mathsf{L}_{\mathrm{ham}}}(h) - \mathcal{R}^*_{\mathsf{L}_{\mathrm{ham}}}(\mathcal{H}_{\mathrm{all}}) \leq 2\left(\mathcal{R}_{\widetilde{\mathsf{L}}_{\log}}(h) - \mathcal{R}^*_{\widetilde{\mathsf{L}}_{\log}}(\mathcal{H}_{\mathrm{all}})\right)^{\frac{1}{2}}. \tag{15}$$

*Moreover, $\widetilde{\mathsf{L}}_{\log}$ is Bayes-consistent with respect to $\mathsf{L}_{\mathrm{ham}}$.*

Approaches like binary relevance surrogate loss treat each label independently. This overlooks crucial inherent information encoded in the relationships between labels. Our new form of surrogate losses explicitly captures these label correlations. Both the binary relevance surrogate loss and our new surrogate loss are Bayes-consistent, meaning that minimizing them over the family of all measurable functions approximates the Bayes-optimal solution. However, our correlation-aware surrogate losses can converge faster, which is reflected in their more favorable $\mathcal{H}$-consistency bounds, independent of the number of labels.

It is known that $\widetilde{\mathsf{L}}_{\mathrm{br}}$ is only Bayes-consistent with respect to the Hamming loss and Precision@$\kappa$, and can be arbitrarily bad for other multi-label losses such as $F_\beta$-measure loss [Dembczynski et al., 2011]. Instead, we will show in the next section that our surrogate loss $\widetilde{\mathsf{L}}_{\log}$ adapts to and is Bayes-consistent with respect to an extensive family of multi-label losses, including the $F_\beta$ measure loss.

## 5 Extension: general multi-label losses

In this section, we broaden our analysis to cover a more extensive family of multi-label losses, including all common ones and a new extension defined based on linear-fractional functions with respect to the confusion matrix. Note that several loss functions are defined over the space $\{0, 1\}^l$, rather than $\{+1, -1\}^l$. To accommodate this difference, any pair $y, y' \in \mathcal{Y} = \{+1, -1\}^l$ can be projected onto $\{0, 1\}^l$ by letting $\overline{y} = \frac{y+1}{2}$ and $\overline{y}' = \frac{y'+1}{2}$, where $\mathbf{1} \in \mathbb{R}^l$ is the vector with all elements equal to 1. Several common multi-label losses are defined as follows.

**Hamming loss:** $\overline{\mathsf{L}}(y', y) = \sum_{i=1}^l \mathbb{1}_{y_i \neq y'_i}$.

**$F_\beta$-measure loss:** $\overline{\mathsf{L}}(y', y) = 1 - \frac{(1+\beta^2)\overline{y}' \cdot \overline{y}}{\beta^2 \|\overline{y}\|_1 + \|\overline{y}'\|_1}$.

**Subset $0/1$ loss:** $\overline{\mathsf{L}}(y', y) = \max_{i \in [l]} \mathbb{1}_{y'_i \neq y_i}$.

**Jaccard distance:** $\overline{\mathsf{L}}(y', y) = 1 - \frac{\overline{y}' \cdot \overline{y}}{\|\overline{y}\|_1 + \|\overline{y}'\|_1 - \overline{y}' \cdot y}$

**Precision@$\kappa$:** $\overline{\mathsf{L}}(y', y) = 1 - \frac{1}{\kappa} \sum_{i \in \mathcal{T}(\overline{y}')} 1_{\overline{y}_i=1}$ subject to $y' \in \mathcal{Y}_\kappa$, where $\mathcal{Y}_\kappa = \{y \in \mathcal{Y} : \|\overline{y}\|_1 = \kappa\}$ and $\mathcal{T}(\overline{y}') = \{i \in [l] : \overline{y}'_i = 1\}$.

**Recall@$\kappa$:** $\overline{\mathsf{L}}(y', y) = 1 - \frac{1}{\|\overline{y}\|_1} \sum_{i \in \mathcal{T}(\overline{y}')} 1_{\overline{y}_i=1}$ subject to $y' \in \mathcal{Y}_\kappa$, where $\mathcal{Y}_\kappa = \{y \in \mathcal{Y} : \|\overline{y}\|_1 = \kappa\}$ and $\mathcal{T}(\overline{y}') = \{i \in [l] : \overline{y}'_i = 1\}$.

More generally, we can define a multi-label loss based on true positives (TP), true negatives (TN), false positives (FP) and false negatives (FN), which can be written explicitly as follows:

$$\mathsf{TP} = \overline{y}' \cdot \overline{y} \quad \mathsf{TN} = \|\overline{y}\|_1 - \overline{y}' \cdot \overline{y}$$
$$\mathsf{FP} = \|\overline{y}'\|_1 - \overline{y}' \cdot \overline{y}, \quad \mathsf{FN} = l + \overline{y}' \cdot \overline{y} - \|\overline{y}\|_1 - \|\overline{y}'\|_1$$

Similar to [Koyejo et al., 2014, 2015], we now define a general family of multi-label losses as linear-fractional functions in terms of these four quantities:

$$\overline{\mathsf{L}}(y', y) = \frac{a_0 + a_{11}\mathsf{TP} + a_{10}\mathsf{FP} + a_{01}\mathsf{FN} + a_{00}\mathsf{TN}}{b_0 + b_{11}\mathsf{TP} + b_{10}\mathsf{FP} + b_{01}\mathsf{FN} + b_{00}\mathsf{TN}}. \tag{16}$$

It can be shown that the aforementioned Hamming loss, $F_\beta$-measure loss, Jaccard distance, precision and recall all belong to this family. Note that the previous definitions in [Koyejo et al., 2014, 2015] are within the empirical utility maximization (EUM) framework [Ye et al., 2012], where the measures are directly defined as functions of the population. We generalize their definition to the decision theoretic analysis (DTA) framework, in terms of loss functions defined over $y$ and $y'$.

Moreover, we can consider extending multi-label losses (16) to non-linear fractional functions of these four quantities, or more generally, to any other forms, as long as they are defined over the space $\mathcal{Y} \times \mathcal{Y}$.

Another important family of multi-label losses is the *tree distance* loss, used in cases of hierarchical classes. In many practical applications, the class labels exist within a predefined hierarchy. For example, in the image tagging problem, class labels might include broad categories such as 'animals' or 'vehicles', which further subdivide into more specific classes like 'mammals' and 'birds' for animals, or 'cars' and 'trucks' for vehicles. Each of these subcategories can be divided even further, showcasing a clear hierarchical structure.

**Tree distance:** Let $T = (\mathcal{Y}, E, W)$ be a tree over the label space $\mathcal{Y}$, with edge set $E$ and positive, finite edge lengths specified by $W$. Suppose $r \in \mathcal{Y}$ is designated as the root node. Then, $\overline{\mathsf{L}}_T(y', y) =$ the shortest path length in $T$ between $y$ and $y'$.

Despite the widespread use of hierarchical classes in practice, to our knowledge, no Bayes-consistent surrogate has been proposed for the tree distance loss in multi-label learning. Next, we will show that our multi-label logistic loss can accommodate all these different loss functions, including the tree distance loss. For any general multi-label loss $\overline{\mathsf{L}}$, we define the multi-label logistic loss as follows:

$$\forall (h, x, y) \in \mathcal{H} \times \mathcal{X} \times \mathcal{Y}, \quad \widetilde{\mathsf{L}}_{\log}(h, x, y) = \sum_{y' \in \mathcal{Y}} \left(1 - \overline{\mathsf{L}}(y', y)\right) \log\left(\sum_{y'' \in \mathcal{Y}} e^{\sum_{i=1}^{l}(y''_i - y'_i)h(x,i)}\right). \tag{17}$$

Here, $\overline{\mathsf{L}}$ can be chosen as all the multi-label losses mentioned above. Next, we will show that $\widetilde{\mathsf{L}}_{\log}$ benefits from $\mathcal{H}$-consistency bounds and Bayes consistency with respect to any of these loss functions.

**Theorem 5.1.** *Let $\mathcal{H} = \mathcal{F}^l$. Assume that $\mathcal{F}$ is complete. Then, the following $\mathcal{H}$-consistency bound holds in the multi-label learning:*

$$\forall h \in \mathcal{H}, \quad \mathcal{R}_{\mathsf{L}}(h) - \mathcal{R}_{\mathsf{L}}^*(\mathcal{H}) + \mathcal{M}_{\mathsf{L}}(\mathcal{H}) \leq 2\left(\mathcal{R}_{\widetilde{\mathsf{L}}_{\log}}(h) - \mathcal{R}_{\widetilde{\mathsf{L}}_{\log}}^*(\mathcal{H}) + \mathcal{M}_{\widetilde{\mathsf{L}}_{\log}}(\mathcal{H})\right)^{\frac{1}{2}}. \tag{18}$$

The proof of Theorem 5.1 is basically the same as that of Theorem 4.1, modulo replacing the Hamming loss $\mathsf{L}_{\text{ham}}$ with a general multi-label loss $\mathsf{L}$. We include it in Appendix B.3 for completeness. When $\mathcal{H} = \mathcal{H}_{\text{all}}$, minimizability gaps $\mathcal{M}_{\widetilde{\mathsf{L}}_{\log}}(\mathcal{H})$ and $\mathcal{M}_{\mathsf{L}}(\mathcal{H})$ vanish, Theorem 4.1 implies excess error bound and Bayes-consistency of multi-label logistic loss with respect to any multi-label loss.

**Corollary 5.2.** *The following excess error bound holds in the multi-label learning: for all $h \in \mathcal{H}_{\text{all}}$,*

$$\mathcal{R}_{\mathsf{L}}(h) - \mathcal{R}_{\mathsf{L}}^*(\mathcal{H}_{\text{all}}) \leq 2\left(\mathcal{R}_{\widetilde{\mathsf{L}}_{\log}}(h) - \mathcal{R}_{\widetilde{\mathsf{L}}_{\log}}^*(\mathcal{H}_{\text{all}})\right)^{\frac{1}{2}}. \tag{19}$$

*Moreover, $\widetilde{\mathsf{L}}_{\log}$ is Bayes-consistent with respect to $\mathsf{L}$.*

Corollary 5.2 is remarkable, as it demonstrates that a unified surrogate loss, $\widetilde{\mathsf{L}}_{\log}$, is Bayes-consistent for any multi-label loss, significantly expanding upon previous work which only established consistency for specific loss functions. Furthermore, Theorem 5.1 provides a stronger guarantee than Bayes-consistency, which is both non-asymptotic and specific to the hypothesis set used.

Minimizing the multi-label logistic loss directly leads to the effective algorithm in multi-label learning. We further discuss the efficiency and practicality of this algorithm in Section 8, where we describe efficient gradient computation.

## 6 Extension: multi-label comp-sum losses

In this section, we further extend our multi-label logistic losses to more comprehensive *multi-label comp-sum losses*, adapting *comp-sum losses* [Mao, Mohri, and Zhong, 2023f] from standard classification to the multi-label learning. As shown by Mao, Mohri, and Zhong [2023f], a comp-sum loss is defined via a composition of the function $\Psi$ and a sum, and includes the logistic loss ($\Psi(u) = \log(u)$) [Verhulst, 1838, 1845, Berkson, 1944, 1951], the *sum-exponential loss* ($\Psi(u) = u - 1$) [Weston and Watkins, 1998, Awasthi et al., 2022b], the *generalized cross-entropy loss* ($\Psi(u) = \frac{1}{q}\left(1 - \frac{1}{u^q}\right), q \in (0,1)$) [Zhang and Sabuncu, 2018], and the *mean absolute error loss* ($\Psi(u) = 1 - \frac{1}{u}$) [Ghosh et al., 2017] as special cases.

Given any multi-label loss $\mathsf{L}$, we will define our novel *multi-label comp-sum losses* as follows:

$$\forall (h, x, y) \in \mathcal{H} \times \mathcal{X} \times \mathcal{Y}, \quad \widetilde{\mathsf{L}}_{\mathrm{comp}}(h, x, y) = \sum_{y' \in \mathcal{Y}} \left(1 - \overline{\mathsf{L}}(y', y)\right) \Psi\left(\sum_{y'' \in \mathcal{Y}} e^{\sum_{i=1}^{l}\left(y_i'' - y_i'\right)h(x,i)}\right). \quad (20)$$

This formulation can be interpreted as a weighted comp-sum loss, where $\left(1 - \overline{\mathsf{L}}(\cdot, y)\right)$ serves as a weight vector. Additionally, this formulation accounts for label correlations among the $y_i$s within the function $\Psi$. Next, we prove that this family of surrogate losses benefits from $\mathcal{H}$-consistency bounds, and thus Bayes-consistency, across any general multi-label loss.

**Theorem 6.1.** *Let $\mathcal{H} = \mathcal{F}^l$. Assume that $\mathcal{F}$ is complete. Then, the following $\mathcal{H}$-consistency bound holds in the multi-label learning:*

$$\forall h \in \mathcal{H}, \quad \mathcal{R}_{\mathsf{L}}(h) - \mathcal{R}_{\mathsf{L}}^*(\mathcal{H}) + \mathcal{M}_{\mathsf{L}}(\mathcal{H}) \leq \Gamma\left(\mathcal{R}_{\widetilde{\mathsf{L}}_{\mathrm{comp}}}(h) - \mathcal{R}_{\widetilde{\mathsf{L}}_{\mathrm{comp}}}^*(\mathcal{H}) + \mathcal{M}_{\widetilde{\mathsf{L}}_{\mathrm{comp}}}(\mathcal{H})\right), \quad (21)$$

*where $\Gamma(t) = 2\sqrt{t}$ when $\Psi(u) = \log(u)$ or $u - 1$; $\Gamma(t) = 2\sqrt{n^q t}$ when $\Psi(u) = \frac{1}{q}\left(1 - \frac{1}{u^q}\right), q \in (0,1)$; and $\Gamma(t) = nt$ when $\Psi(u) = 1 - \frac{1}{u}$.*

**Corollary 6.2.** *The following excess error bound holds in the multi-label learning:*

$$\forall h \in \mathcal{H}_{\mathrm{all}}, \quad \mathcal{R}_{\mathsf{L}}(h) - \mathcal{R}_{\mathsf{L}}^*(\mathcal{H}_{\mathrm{all}}) \leq \Gamma\left(\mathcal{R}_{\widetilde{\mathsf{L}}_{\mathrm{comp}}}(h) - \mathcal{R}_{\widetilde{\mathsf{L}}_{\mathrm{comp}}}^*(\mathcal{H}_{\mathrm{all}})\right), \quad (22)$$

*where $\Gamma(t) = 2\sqrt{t}$ when $\Psi(u) = \log(u)$ or $u - 1$; $\Gamma(t) = 2\sqrt{n^q t}$ when $\Psi(u) = \frac{1}{q}\left(1 - \frac{1}{u^q}\right), q \in (0,1)$; and $\Gamma(t) = nt$ when $\Psi(u) = 1 - \frac{1}{u}$. Moreover, $\widetilde{\mathsf{L}}_{\mathrm{comp}}$ with these choices of $\Psi$ are Bayes-consistent with respect to $\mathsf{L}$.*

The proof of Theorem 6.1 is included in Appendix B.4. Similar to the proof of Theorem 5.1, we make use of Theorem B.1 and Lemma B.2 in Appendix B.1. However, upper bounding the conditional regret of $\mathsf{L}$ by that of the surrogate loss $\widetilde{\mathsf{L}}_{\mathrm{comp}}$ for different choices of $\Psi$ requires a distinct analysis depending on the specific form of the function $\Psi$, leading to various concave functions $\Gamma$. Our proof is inspired by the proof of $\mathcal{H}$-consistency bounds for comp-sum losses in [Mao et al., 2023f] through the introduction of a parameter $\mu$ and optimization. However, the novelty lies in the adaptation of $\mu$ with a quantity $\mathsf{s}$ tailored to multi-label loss functions instead of the score vector $h$ itself.

Note that, as with $\Psi(u) = \log(u)$ shown in Section 5, for $\Psi(u) = u - 1$, the bounds are also independent of the number of labels and are favorable. However, for other choices of $\Psi$, the bounds exhibit a worse dependency on $n$, which can be exponential with respect to $l$.

## 7 Extension: multi-label constrained losses

In this section, we introduce another novel family of surrogate losses, adapting *constrained losses* [Lee et al., 2004, Awasthi et al., 2022b] from standard classification to *multi-label constrained losses*

in a similar way. Given any general multi-label loss $\mathsf{L}$, we define *multi-label constrained losses* as:

$$\forall (h, x, y) \in \mathcal{H} \times \mathcal{X} \times \mathcal{Y}, \quad \widetilde{\mathsf{L}}_{\text{cstnd}}(h, x, y) = \sum_{y' \in \mathcal{Y}} \overline{\mathsf{L}}(y', y) \Phi\left(-\sum_{i=1}^{l} y_i' h(x, i)\right). \tag{23}$$

where $\sum_{y \in \mathcal{Y}} \sum_{i=1}^{l} y_i h(x, i) = 0$. Next, we show that $\widetilde{\mathsf{L}}_{\text{cstnd}}$ also benefit from $\mathcal{H}$-consistency bounds and thus Bayes-consistency for any multi-label loss.

**Theorem 7.1.** *Let $\mathcal{H} = \mathcal{F}^l$. Assume that $\mathcal{F}$ is complete Then, the following $\mathcal{H}$-consistency bound holds in the multi-label learning:*

$$\forall h \in \mathcal{H}, \quad \mathcal{R}_{\mathsf{L}}(h) - \mathcal{R}_{\mathsf{L}}^*(\mathcal{H}) + \mathcal{M}_{\mathsf{L}}(\mathcal{H}) \leq \Gamma\left(\mathcal{R}_{\widetilde{\mathsf{L}}_{\text{cstnd}}}(h) - \mathcal{R}_{\widetilde{\mathsf{L}}_{\text{cstnd}}}^*(\mathcal{H}) + \mathcal{M}_{\widetilde{\mathsf{L}}_{\text{cstnd}}}(\mathcal{H})\right), \tag{24}$$

*where $\Gamma(t) = 2\sqrt{\mathsf{L}_{\max} t}$ when $\Phi(u) = e^{-u}$; $\Gamma(t) = 2\sqrt{t}$ when $\Phi(u) = \max\{0, 1-u\}^2$; and $\Gamma(t) = t$ when $\Phi(u) = \max\{0, 1-u\}$ or $\Phi(u) = \min\{\max\{0, 1-u/\rho\}, 1\}$, $\rho > 0$.*

**Corollary 7.2.** *The following excess error bound holds in the multi-label learning:*

$$\forall h \in \mathcal{H}_{\text{all}}, \quad \mathcal{R}_{\mathsf{L}}(h) - \mathcal{R}_{\mathsf{L}}^*(\mathcal{H}_{\text{all}}) \leq \Gamma\left(\mathcal{R}_{\widetilde{\mathsf{L}}_{\text{cstnd}}}(h) - \mathcal{R}_{\widetilde{\mathsf{L}}_{\text{cstnd}}}^*(\mathcal{H}_{\text{all}})\right), \tag{25}$$

*where $\Gamma(t) = 2\sqrt{\mathsf{L}_{\max} t}$ when $\Phi(u) = e^{-u}$; $\Gamma(t) = 2\sqrt{t}$ when $\Phi(u) = \max\{0, 1-u\}^2$; and $\Gamma(t) = t$ when $\Phi(u) = \max\{0, 1-u\}$ or $\Phi(u) = \min\{\max\{0, 1-u/\rho\}, 1\}$, $\rho > 0$. Moreover, $\widetilde{\mathsf{L}}_{\text{cstnd}}$ with these choices of $\Phi$ are Bayes-consistent with respect to $\mathsf{L}$.*

The proof of Theorem 7.1 is included in Appendix B.5. As with the proof of Theorem 6.1, we use Theorem B.1 and Lemma B.2 from Appendix B.1, and aim to upper bound the conditional regret of $\mathsf{L}$ by that of the surrogate losses $\widetilde{\mathsf{L}}_{\text{comp}}$ using various concave functions $\Gamma$. However, the difference lies in our introduction and optimization of a parameter $\mu$ tailored to a quantity $z$ that is specific to the form of the multi-label constrained loss.

These results show that in cases where minimizability gaps vanish, reducing the estimation error of $\widetilde{\mathsf{L}}_{\text{cstnd}}$ to $\epsilon$ results in the estimation error of target multi-label loss $\mathsf{L}$ being upper bounded by either $\sqrt{\epsilon}$ or $\epsilon$, modulo a constant that is independent of the number of labels.

# 8  Efficient Gradient Computation

In this section, we demonstrate the efficient computation of the gradient for the multi-label logistic loss $\widetilde{\mathsf{L}}_{\log}$ at any point $(x^j, y^j)$. This loss function is therefore both theoretically grounded in $\mathcal{H}$-consistency bounds and computationally efficient. Consider the labeled pair $(x^j, y^j)$ and a hypothesis $h$ in $\mathcal{H}$. The expression for $\widetilde{\mathsf{L}}_{\log}(h, x^j, y^j)$ can be reformulated as follows:

$$\widetilde{\mathsf{L}}_{\log}(h, x^j, y^j) = \sum_{y' \in \mathcal{Y}} \left(1 - \overline{\mathsf{L}}(y', y^j)\right) \log\left(\sum_{y'' \in \mathcal{Y}} e^{\sum_{i=1}^{l}(y_i'' - y_i')h(x^j, i)}\right)$$

$$= -\sum_{y' \in \mathcal{Y}} \left(1 - \overline{\mathsf{L}}(y', y^j)\right) \sum_{i=1}^{l} y_i' h(x^j, i) + \sum_{y' \in \mathcal{Y}} \left(1 - \overline{\mathsf{L}}(y', y^j)\right) \log\left(\sum_{y \in \mathcal{Y}} e^{\sum_{i=1}^{l} y_i h(x^j, i)}\right).$$

Let $\mathsf{L}_1(j) = \sum_{y \in \mathcal{Y}}\left(1 - \overline{\mathsf{L}}(y, y^j)\right)$, which is independent of $h$ and can be pre-computed. It can also be invariant with respect to $j$ and is a fixed constant for many loss functions such as the Hamming loss.

For many commonly used loss functions, the terms involving sums over all possible label combinations can be simplified analytically. To illustrate this, we provide explicit formulae for the Hamming loss and $F_\beta$ measure loss functions:

**Hamming loss**: $\sum_{y \in \mathcal{Y}}\left(1 - \overline{\mathsf{L}}_{\text{ham}}(y, y^j)\right) = -(l-1)2^l + \sum_{y \in \mathcal{Y}}\left(l - \overline{\mathsf{L}}_{\text{ham}}(y, y^j)\right) = -(l-1)2^l + \sum_{y \in \mathcal{Y}} \sum_{i=1}^{l} \mathbb{1}_{y_i = y_i^j} = -(l-1)2^l + \sum_{i=1}^{l} \sum_{y \in \mathcal{Y}} \mathbb{1}_{y_i = y_i^j} = -(l-1)2^l + 2^{l-1}l = 2^{l-1}(2-l)$.

$F_\beta$ **measure**: $\sum_{y \in \mathcal{Y}}\left(1 - \overline{\mathsf{L}}_{F_\beta}(y, y^j)\right) = \sum_{y \in \mathcal{Y}} \frac{(1+\beta^2)\overline{y} \cdot \overline{y}^j}{\beta^2 \|\overline{y}\|_1 + \|\overline{y}^j\|_1} = \sum_{k=0}^{n} \sum_{y \in G_k} \frac{(1+\beta^2)\overline{y} \cdot \overline{y}^j}{\beta^2 k + \|\overline{y}^j\|_1} = \sum_{k=0}^{n} \frac{(1+\beta^2)\sum_{i=1}^{l} y_i^j \binom{n-1}{k-1}}{\beta^2 k + \|\overline{y}^j\|_1} = \sum_{k=0}^{l} \frac{(1+\beta^2)\|\overline{y}^j\|_1 \binom{l-1}{k-1}}{\beta^2 k + \|\overline{y}^j\|_1}$, where $G_k = \{y \in \mathcal{Y} : \|\overline{y}\|_1 = k\}$.

A similar analysis applies to many other loss functions, thus, these terms do not affect the tractability of our algorithms. Next, we will consider the hypothesis set of linear functions $\mathcal{H} = \left\{ x \mapsto \mathbf{w} \cdot \Psi(x, i) \colon \mathbf{w} \in \mathbb{R}^d \right\}$, where $\Psi$ is a feature mapping from $\mathcal{X} \times [l]$ to $\mathbb{R}^d$. Using the shorthand $\mathbf{w}$ for $h$, we can rewrite $\widetilde{\mathsf{L}}_{\log}$ at $(x^j, y^j)$ as follows:

$$\widetilde{\mathsf{L}}_{\log}(\mathbf{w}, x^j, y^j) = -\mathbf{w} \cdot \left[ \sum_{y' \in \mathcal{Y}} \left(1 - \overline{\mathsf{L}}(y', y^j)\right) \left( \sum_{i=1}^{l} y_i' \Psi(x^j, i) \right) \right] + \mathsf{L}_1(j) \log(Z_{\mathbf{w}, j}), \qquad (26)$$

where $Z_{\mathbf{w}, j} = \sum_{y \in \mathcal{Y}} e^{\mathbf{w} \cdot \left( \sum_{i=1}^{l} y_i \Psi(x^j, i) \right)}$. Then, we can compute the gradient of $\widetilde{\mathsf{L}}_{\log}$ at any $\mathbf{w} \in \mathbb{R}^d$:

$$\nabla \widetilde{\mathsf{L}}_{\log}(\mathbf{w}) = - \sum_{y' \in \mathcal{Y}} \left(1 - \overline{\mathsf{L}}(y', y^j)\right) \left( \sum_{i=1}^{l} y_i' \Psi(x^j, i) \right) + \mathsf{L}_1(j) \sum_{y \in \mathcal{Y}} \frac{e^{\mathbf{w} \cdot \left( \sum_{i=1}^{l} y_i \Psi(x^j, i) \right)}}{Z_{\mathbf{w}, j}} \left( \sum_{i=1}^{l} y_i \Psi(x^j, i) \right)$$

$$= - \sum_{y' \in \mathcal{Y}} \left(1 - \overline{\mathsf{L}}(y', y^j)\right) \left( \sum_{i=1}^{l} y_i' \Psi(x^j, i) \right) + \mathsf{L}_1(j) \, \mathop{\mathbb{E}}_{y \sim \mathsf{q}_{\mathbf{w}}} \left[ \left( \sum_{i=1}^{l} y_i \Psi(x^j, i) \right) \right], \qquad (27)$$

where $\mathsf{q}_{\mathbf{w}}$ is a distribution over $\mathcal{Y}$ with probability mass function $\mathsf{q}_{\mathbf{w}}(y) = \frac{e^{\mathbf{w} \cdot \left( \sum_{i=1}^{l} y_i \Psi(x^j, i) \right)}}{Z_{\mathbf{w}, j}}$. By rearranging the terms in (27), we obtain the following result.

**Lemma 8.1.** *The gradient of $\widetilde{\mathsf{L}}_{\log}$ at any $\mathbf{w} \in \mathbb{R}^d$ can be expressed as follows:*

$$\nabla \widetilde{\mathsf{L}}_{\log}(\mathbf{w}) = \sum_{i=1}^{l} \Psi(x^j, i) \mathsf{L}_2(i, j) + \mathsf{L}_1(j) \sum_{i=1}^{l} \Psi(x^j, i) Q_{\mathbf{w}}(i) \qquad (28)$$

*where $\mathsf{L}_2(i, j) = \sum_{y \in \mathcal{Y}} \left(1 - \overline{\mathsf{L}}(y, y^j)\right) y_i$, $\mathsf{L}_1(j) = \sum_{y \in \mathcal{Y}} \left(1 - \overline{\mathsf{L}}(y, y^j)\right)$, $Q_{\mathbf{w}}(i) = \sum_{y \in \mathcal{Y}} \mathsf{q}_{\mathbf{w}}(y) y_i$, $\mathsf{q}_{\mathbf{w}}(y) = \frac{e^{\mathbf{w} \cdot \left( \sum_{i=1}^{l} y_i \Psi(x^j, i) \right)}}{Z_{\mathbf{w}, j}}$, and $Z_{\mathbf{w}, j} = \sum_{y \in \mathcal{Y}} e^{\mathbf{w} \cdot \left( \sum_{i=1}^{l} y_i \Psi(x^j, i) \right)}$. The overall time complexity for gradient computation is $O(l)$.*

Here, the evaluation of $\mathsf{L}_2(i, j)$, $i \in [l]$ and $\mathsf{L}_1(j)$ can be computed once and for all, before any gradient computation. For evaluation of $Q_{\mathbf{w}}(i)$, note that it can be equivalently written as follows:

$$Q_{\mathbf{w}}(i) = \sum_{y \in \mathcal{Y}} \frac{e^{\mathbf{w} \cdot \widetilde{\Psi}(x^j, y)}}{\sum_{y \in \mathcal{Y}} e^{\mathbf{w} \cdot \widetilde{\Psi}(x^j, y)}} y_i, \text{ with } \widetilde{\Psi}(x^j, y) = \sum_{i=1}^{l} y_i \Psi(x^j, i), \qquad (29)$$

where $\widetilde{\Psi}(x^j, y)$ admits a *Markovian property of order* 1 [Manning and Schutze, 1999, Cortes et al., 2016]. Thus, as shown by Cortes et al. [2016, 2018], $Q_{\mathbf{w}}(i)$ can be evaluated efficiently by running two single-source shortest-distance algorithms over the $(+, \times)$ semiring on an appropriate weighted finite automaton (WFA). More specifically, in our case, the WFA can be described as follows: there are $(l + 1)$ vertices labeled $0, \ldots, l$. There are two transitions from $k$ to $(k + 1)$ labeled with $+1$ and $-1$. The weight of the transition with label $+1$ is $\exp(+\mathbf{w} \cdot \widetilde{\Psi}(x^j, k))$, and $\exp(-\mathbf{w} \cdot \widetilde{\Psi}(x^j, k))$ for the other. $0$ is the initial state, and $l$ the final state. The overall time complexity of computing all quantities $Q_{\mathbf{w}}(i)$, $i \in [l]$, is $O(l)$.

## 9 Conclusion

We presented a comprehensive analysis of surrogate losses for multi-label learning, establishing strong consistency guarantees. We introduced a novel multi-label logistic loss that addresses the shortcomings of existing methods and enjoys label-independent consistency bounds. Our proposed family of multi-label comp-sum losses offers a unified framework with strong consistency guarantees for any general multi-label loss, significantly expanding upon previous work. Additionally, we presented efficient algorithms for their gradient computation. While empirical validation is left for future work, our theoretical results demonstrate the potential of these new surrogate losses to advance multi-label learning. This unified framework shows promise for broader applications and paves the way for future research in multi-label learning and related areas.

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

# Contents of Appendix

# A  Proof of $\mathcal{H}$-consistency bounds for existing surrogate losses (Theorem 3.1)

**Theorem 3.1.** *Let $\mathcal{H} = \mathcal{F}^l$. Assume that the following $\mathcal{F}$-consistency bound holds in the binary classification, for some concave function $\Gamma \colon \mathbb{R} \to \mathbb{R}_+$:*

$$\forall f \in \mathcal{F}, \quad \mathcal{R}_{\ell_{0-1}}(f) - \mathcal{R}^*_{\ell_{0-1}}(\mathcal{F}) + \mathcal{M}_{\ell_{0-1}}(\mathcal{F}) \le \Gamma(\mathcal{R}_\Phi(f) - \mathcal{R}^*_\Phi(\mathcal{F}) + \mathcal{M}_\Phi(\mathcal{F})). \tag{8}$$

*Then, the following $\mathcal{H}$-consistency bound holds in the multi-label learning: for all $h \in \mathcal{H}$,*

$$\mathcal{R}_{\mathsf{L}_{\mathrm{ham}}}(h) - \mathcal{R}^*_{\mathsf{L}_{\mathrm{ham}}}(\mathcal{H}) + \mathcal{M}_{\mathsf{L}_{\mathrm{ham}}}(\mathcal{H}) \le l\Gamma\left(\frac{\mathcal{R}_{\widetilde{\mathsf{L}}_{\mathrm{br}}}(h) - \mathcal{R}^*_{\widetilde{\mathsf{L}}_{\mathrm{br}}}(\mathcal{H}) + \mathcal{M}_{\widetilde{\mathsf{L}}_{\mathrm{br}}}(\mathcal{H})}{l}\right). \tag{9}$$

*Proof.* Let $p(y \mid x) = \mathbb{P}(Y = y \mid X = x)$ be the conditional probability of $Y = y$ given $X = x$. Given a multi-label surrogate loss $\widetilde{\mathsf{L}}$ and a hypothesis set $\mathcal{H}$, we denote the conditional error by $\mathcal{C}_{\widetilde{\mathsf{L}}}(h, x) = \mathbb{E}_{y|x}\big[\widetilde{\mathsf{L}}(h, x, y)\big]$, the best-in-class conditional error by $\mathcal{C}^*_{\widetilde{\mathsf{L}}}(\mathcal{H}, x) = \inf_{h \in \mathcal{H}} \mathcal{C}_{\widetilde{\mathsf{L}}}(h, x)$, and the conditional regret by $\Delta\mathcal{C}_{\widetilde{\mathsf{L}}, \mathcal{H}}(h, x) = \mathcal{C}_{\widetilde{\mathsf{L}}}(h, x) - \mathcal{C}^*_{\widetilde{\mathsf{L}}}(\mathcal{H}, x)$. We can express the conditional error of the hamming loss and the surrogate loss $\widetilde{\mathsf{L}}_{\mathrm{br}}$ as follows:

$$\begin{aligned}
\mathcal{C}_{\mathsf{L}_{\mathrm{ham}}}(h, x) &= \sum_{y \in \mathcal{Y}} p(y \mid x) \sum_{i=1}^{l} 1_{y_i \ne h(x, i)} \\
&= \sum_{i=1}^{l}\left(\sum_{y:y_i=+1} p(y \mid x) 1_{1 \ne \mathrm{sign}(h(x,i))} + \sum_{y:y_i=-1} p(y \mid x) 1_{-1 \ne \mathrm{sign}(h(x,i))}\right) \\
\mathcal{C}_{\widetilde{\mathsf{L}}_{\mathrm{br}}}(h, x) &= \sum_{y \in \mathcal{Y}} p(y \mid x) \sum_{i=1}^{l} \Phi(y_i h(x, i)) \\
&= \sum_{i=1}^{l}\left(\sum_{y:y_i=+1} p(y \mid x) \Phi(h(x, i)) + \sum_{y:y_i=-1} p(y \mid x) \Phi(-h(x, i))\right)
\end{aligned}$$

Let $q_i(+1 \mid x) = \sum_{y:y_i=+1} p(y \mid x)$ and $q_i(-1 \mid x) = \sum_{y:y_i=-1} p(y \mid x)$. Let $f_i = h(\cdot, i) \in \mathcal{F}$, for all $i \in [l]$. Then, it is clear that the conditional regrets of $\mathsf{L}_{\mathrm{ham}}$ and $\widetilde{\mathsf{L}}_{\mathrm{br}}$ can be expressed as the corresponding conditional regrets of $\ell_{0-1}$ and $\Phi$ under this new introduced new distribution:

$$\Delta\mathcal{C}_{\mathsf{L}_{\mathrm{ham}}, \mathcal{H}}(h, x) = \sum_{i=1}^{l} \Delta\mathcal{C}_{\ell_{0-1}, \mathcal{F}}(f_i, x), \quad \Delta\mathcal{C}_{\widetilde{\mathsf{L}}_{\mathrm{br}}, \mathcal{H}}(h, x) = \sum_{i=1}^{l} \Delta\mathcal{C}_{\Phi, \mathcal{F}}(f_i, x). \tag{30}$$

Since we have $\Delta\mathcal{C}_{\ell_{0-1}, \mathcal{F}}(f_i, x) \le \Gamma(\Delta\mathcal{C}_{\Phi, \mathcal{F}}(f_i, x))$ under the assumption, we obtain

$$\begin{aligned}
\Delta\mathcal{C}_{\mathsf{L}_{\mathrm{ham}}, \mathcal{H}}(h, x) = \sum_{i=1}^{l} \Delta\mathcal{C}_{\ell_{0-1}, \mathcal{F}}(f_i, x) &\le \sum_{i=1}^{l} \Gamma(\Delta\mathcal{C}_{\Phi, \mathcal{F}}(f_i, x)) \\
&\le l\Gamma\left(\frac{1}{l}\sum_{i=1}^{l} \Delta\mathcal{C}_{\Phi, \mathcal{F}}(f_i, x)\right) && \text{(concavity of } \Gamma) \\
&= l\Gamma\left(\frac{1}{l}\Delta\mathcal{C}_{\widetilde{\mathsf{L}}_{\mathrm{br}}, \mathcal{H}}(h, x)\right).
\end{aligned}$$

By taking the expectation on both sides and using the Jensen's inequality, we have

$$\begin{aligned}
\mathcal{R}_{\mathsf{L}_{\mathrm{ham}}}(h) - \mathcal{R}^*_{\mathsf{L}_{\mathrm{ham}}}(\mathcal{H}) + \mathcal{M}_{\mathsf{L}_{\mathrm{ham}}}(\mathcal{H}) &= \mathbb{E}_x[\Delta\mathcal{C}_{\mathsf{L}_{\mathrm{ham}}, \mathcal{H}}(h, x)] \\
&\le \mathbb{E}_x\left[l\Gamma\left(\frac{1}{l}\Delta\mathcal{C}_{\widetilde{\mathsf{L}}_{\mathrm{br}}, \mathcal{H}}(h, x)\right)\right] \\
&\le l\Gamma\left(\frac{\mathbb{E}_x[\Delta\mathcal{C}_{\widetilde{\mathsf{L}}_{\mathrm{br}}, \mathcal{H}}(h, x)]}{l}\right) && \text{(concavity of } \Gamma) \\
&= l\Gamma\left(\frac{\mathcal{R}_{\widetilde{\mathsf{L}}_{\mathrm{br}}}(h) - \mathcal{R}^*_{\widetilde{\mathsf{L}}_{\mathrm{br}}}(\mathcal{H}) + \mathcal{M}_{\widetilde{\mathsf{L}}_{\mathrm{br}}}(\mathcal{H})}{l}\right).
\end{aligned}$$

This completes the proof. $\qquad\square$

# B   Proofs of $\mathcal{H}$-consistency bounds for new surrogate losses

## B.1   Auxiliary definitions and results (Theorem B.1 and Lemma B.2)

Before proceeding with the proof, we first introduce some notation and definitions. Given a multi-label surrogate loss $\widetilde{\mathsf{L}}$ and a hypothesis set $\mathcal{H}$, we denote the conditional error by $\mathcal{C}_{\widetilde{\mathsf{L}}}(h, x) = \mathbb{E}_{y|x}\big[\widetilde{\mathsf{L}}(h, x, y)\big]$, the best-in-class conditional error by $\mathcal{C}_{\widetilde{\mathsf{L}}}^*(\mathcal{H}, x) = \inf_{h \in \mathcal{H}} \mathcal{C}_{\widetilde{\mathsf{L}}}(h, x)$, and the conditional regret by $\Delta\mathcal{C}_{\widetilde{\mathsf{L}}, \mathcal{H}}(h, x) = \mathcal{C}_{\widetilde{\mathsf{L}}}(h, x) - \mathcal{C}_{\widetilde{\mathsf{L}}}^*(\mathcal{H}, x)$. We then present a general theorem, which shows that to derive $\mathcal{H}$-consistency bounds in multi-label learning with a concave function $\Gamma$, it is only necessary to upper bound the conditional regret of the target multi-label loss by that of the surrogate loss with the same $\Gamma$.

**Theorem B.1.** *Let* $\mathsf{L}$ *be a multi-label loss and* $\widetilde{\mathsf{L}}$ *be a surrogate loss. Given a concave function* $\Gamma \colon \mathbb{R}_+ \to \mathbb{R}_+$. *If the following condition holds for all* $h \in \mathcal{H}$ *and* $x \in \mathcal{X}$:

$$\Delta\mathcal{C}_{\mathsf{L}, \mathcal{H}}(h, x) \le \Gamma\big(\Delta\mathcal{C}_{\widetilde{\mathsf{L}}, \mathcal{H}}(h, x)\big), \tag{31}$$

*then, for any distribution and for all hypotheses* $h \in \mathcal{H}$,

$$\mathcal{R}_{\mathsf{L}}(h) - \mathcal{R}_{\mathsf{L}}^*(\mathcal{H}) + \mathcal{M}_{\mathsf{L}}(\mathcal{H}) \le \Gamma\big(\mathcal{R}_{\widetilde{\mathsf{L}}}(h) - \mathcal{R}_{\widetilde{\mathsf{L}}}^*(\mathcal{H}) + \mathcal{M}_{\widetilde{\mathsf{L}}}(\mathcal{H})\big). \tag{32}$$

*Proof.* By the definitions, the expectation of the conditional regrets for $\mathsf{L}$ and $\widetilde{\mathsf{L}}$ can be expressed as:

$$\mathbb{E}_x\big[\Delta\mathcal{C}_{\mathsf{L}, \mathcal{H}}(h, x)\big] = \mathcal{R}_{\mathsf{L}}(h) - \mathcal{R}_{\mathsf{L}}^*(\mathcal{H}) + \mathcal{M}_{\mathsf{L}}(\mathcal{H})$$

$$\mathbb{E}_x\big[\Delta\mathcal{C}_{\widetilde{\mathsf{L}}, \mathcal{H}}(h, x)\big] = \mathcal{R}_{\widetilde{\mathsf{L}}}(h) - \mathcal{R}_{\widetilde{\mathsf{L}}}^*(\mathcal{H}) + \mathcal{M}_{\widetilde{\mathsf{L}}}(\mathcal{H}).$$

Thus, by taking the expectation on both sides of (31) and using Jensen's inequality, we have

$$\begin{aligned}
\mathcal{R}_{\mathsf{L}}(h) - \mathcal{R}_{\mathsf{L}}^*(\mathcal{H}) + \mathcal{M}_{\mathsf{L}}(\mathcal{H}) &= \mathbb{E}_x\big[\Delta\mathcal{C}_{\mathsf{L}, \mathcal{H}}(h, x)\big] \\
&\le \mathbb{E}_x\big[\Gamma\big(\Delta\mathcal{C}_{\widetilde{\mathsf{L}}, \mathcal{H}}(h, x)\big)\big] && \text{(Eq. (31))} \\
&\le \Gamma\Big(\mathbb{E}_x\big[\Delta\mathcal{C}_{\widetilde{\mathsf{L}}, \mathcal{H}}(h, x)\big]\Big) && \text{(concavity of } \Gamma) \\
&= \Gamma\big(\mathcal{R}_{\widetilde{\mathsf{L}}}(h) - \mathcal{R}_{\widetilde{\mathsf{L}}}^*(\mathcal{H}) + \mathcal{M}_{\widetilde{\mathsf{L}}}(\mathcal{H})\big).
\end{aligned}$$

This completes the proof.  □

To derive $\mathcal{H}$-consistency bounds using Theorem B.1, we will characterize the conditional regret of a multi-label loss $\mathsf{L}$. For simplicity, we first introduce some notation. For any $x \in \mathcal{X}$, let $\mathsf{y}(x) = \operatorname{argmin}_{y' \in \mathcal{Y}} \mathbb{E}_{y|x}\big[\overline{\mathsf{L}}(y', y)\big] \in \mathcal{Y}$, where we choose the label with the lowest index under the natural ordering of labels as the tie-breaking strategy. To simplify the notation further, we will drop the dependency on $x$. Specifically, we use $\mathsf{y}$ to denote $\mathsf{y}(x)$ and $\mathsf{h}$ to denote $\mathsf{h}(x)$. Additionally, we define $\mathsf{c}_{\mathsf{h}} = \mathbb{E}_{y|x}\big[\overline{\mathsf{L}}(\mathsf{h}, y)\big]$, $\mathsf{c}_{\mathsf{y}} = \mathbb{E}_{y|x}\big[\overline{\mathsf{L}}(\mathsf{y}, y)\big]$ and $\mathsf{c}_{y'} = \mathbb{E}_{y|x}\big[\overline{\mathsf{L}}(y', y)\big]$, $\forall y' \in \mathcal{Y}$.

**Lemma B.2.** *Let* $\mathcal{H} = \mathcal{F}^l$. *Assume that* $\mathcal{F}$ *is complete. Then, the conditional regret of a multi-label loss* $\mathsf{L}$ *can be expressed as follows:* $\Delta\mathcal{C}_{\mathsf{L}, \mathcal{H}}(h, x) = \mathsf{c}_{\mathsf{h}} - \mathsf{c}_{\mathsf{y}}$.

*Proof.* By definition, the conditional error of $\mathsf{L}$ can be expressed as follows:

$$\mathcal{C}_{\mathsf{L}}(h, x) = \mathbb{E}_{y|x}\big[\mathsf{L}(h, x, y)\big] = \mathbb{E}_{y|x}\big[\overline{\mathsf{L}}(\mathsf{h}(x), y)\big] = \mathsf{c}_{\mathsf{h}}. \tag{33}$$

Since $\mathcal{H} = \mathcal{F}^l$ and $\mathcal{F}$ is complete, for any $x \in \mathcal{X}$, $\{\mathsf{h}(x) \colon h \in \mathcal{H}\} = \mathcal{Y}$. Then, the best-in-class conditional error of $\mathsf{L}$ can be expressed as follows:

$$\mathcal{C}_{\mathsf{L}}^*(\mathcal{H}, x) = \inf_{h \in \mathcal{H}} \mathcal{C}_{\mathsf{L}}(h, x) = \inf_{h \in \mathcal{H}} \mathbb{E}_{y|x}\big[\overline{\mathsf{L}}(\mathsf{h}(x), y)\big] = \mathbb{E}_{y|x}\big[\overline{\mathsf{L}}(\mathsf{y}(x), y)\big] = \mathsf{c}_{\mathsf{y}}. \tag{34}$$

Therefore, $\Delta\mathcal{C}_{\mathsf{L}, \mathcal{H}}(h, x) = \mathcal{C}_{\mathsf{L}}(h, x) - \mathcal{C}_{\mathsf{L}}^*(\mathcal{H}, x) = \mathsf{c}_{\mathsf{h}} - \mathsf{c}_{\mathsf{y}}$.  □

Next, by using Lemma B.2, we will upper bound the conditional regret of the target multi-label loss $\mathsf{L}$ by that of the surrogate loss $\widetilde{\mathsf{L}}$ with a concave function $\Gamma$.

## B.2 Proof of Theorem 4.1

**Theorem 4.1.** *Let $\mathcal{H} = \mathcal{F}^l$. Assume that $\mathcal{F}$ is complete. Then, the following $\mathcal{H}$-consistency bound holds in the multi-label learning: for all $h \in \mathcal{H}$,*

$$\mathcal{R}_{\mathsf{L}_{\mathrm{ham}}}(h) - \mathcal{R}^*_{\mathsf{L}_{\mathrm{ham}}}(\mathcal{H}) + \mathcal{M}_{\mathsf{L}_{\mathrm{ham}}}(\mathcal{H}) \le 2\Big(\mathcal{R}_{\widetilde{\mathsf{L}}_{\mathrm{log}}}(h) - \mathcal{R}^*_{\widetilde{\mathsf{L}}_{\mathrm{log}}}(\mathcal{H}) + \mathcal{M}_{\widetilde{\mathsf{L}}_{\mathrm{log}}}(\mathcal{H})\Big)^{\frac{1}{2}}. \tag{13}$$

*Proof.* We will use the following notation adapted to the Hamming loss: $\mathsf{c}_{\mathsf{h}} = \mathbb{E}_{y|x}\big[\overline{\mathsf{L}}_{\mathrm{ham}}(\mathsf{h}, y)\big]$, $\mathsf{c}_{\mathsf{y}} = \mathbb{E}_{y|x}\big[\overline{\mathsf{L}}(\mathsf{y}, y)\big]$ and $\mathsf{c}_{y'} = \mathbb{E}_{y|x}\big[\overline{\mathsf{L}}_{\mathrm{ham}}(y', y)\big]$, $\forall y' \in \mathcal{Y}$. We will denote by $\mathsf{s}(h, x, y') = \frac{e^{\sum_{i=1}^{l} y'_i h(x,i)}}{\sum_{y'' \in \mathcal{Y}} e^{\sum_{i=1}^{l} y''_i h(x,i)}}$ and simplify notation by using $\mathsf{s}_{y'}$, thereby dropping the dependency on $h$ and $x$. It is clear that $\mathsf{s}_{y'} \in [0, 1]$. Then, the conditional error of $\widetilde{\mathsf{L}}_{\mathrm{log}}$ can be expressed as follows:

$$\mathcal{C}_{\widetilde{\mathsf{L}}_{\mathrm{log}}}(h, x) = \mathbb{E}_{y|x}\Bigg[\sum_{y' \in \mathcal{Y}}\big(1 - \overline{\mathsf{L}}_{\mathrm{ham}}(y', y)\big) \log\Bigg(\sum_{y'' \in \mathcal{Y}} e^{\sum_{i=1}^{l}(y''_i - y'_i)h(x,i)}\Bigg)\Bigg]$$

$$= -\sum_{y' \in \mathcal{Y}}\big(1 - \mathsf{c}_{y'}\big) \log(\mathsf{s}_{y'})$$

For any $\mathsf{h} \ne \mathsf{y}$, we define $\mathsf{s}^\mu$ as follows: set $\mathsf{s}^\mu_{y'} = \mathsf{s}_{y'}$ for all $y' \ne \mathsf{y}$ and $y' \ne \mathsf{h}$; define $\mathsf{s}^\mu_{\mathsf{h}} = \mathsf{s}_{\mathsf{y}} - \mu$; and let $\mathsf{s}^\mu_{\mathsf{y}} = \mathsf{s}_{\mathsf{h}} + \mu$. Note that $\mathsf{s}^\mu$ can be realized by some $h' \in \mathcal{H}$ due to the completeness assumption. Then, we have

$$\Delta\mathcal{C}_{\widetilde{\mathsf{L}}_{\mathrm{log}}, \mathcal{H}}(h, x) \ge \Bigg(-\sum_{y' \in \mathcal{Y}}\big(1 - \mathsf{c}_{y'}\big) \log(\mathsf{s}_{y'})\Bigg) - \inf_{\mu \in \mathbb{R}}\Bigg(-\sum_{y' \in \mathcal{Y}}\big(1 - \mathsf{c}_{y'}\big) \log(\mathsf{s}^\mu_{y'})\Bigg)$$

$$= \sup_{\mu \in \mathbb{R}}\big\{(1 - \mathsf{c}_{\mathsf{h}})[\log(\mathsf{s}_{\mathsf{y}} - \mu) - \log(\mathsf{s}_{\mathsf{h}})] + (1 - \mathsf{c}_{\mathsf{y}})[\log(\mathsf{s}_{\mathsf{h}} + \mu) - \log(\mathsf{s}_{\mathsf{y}})]\big\}$$

$$= (1 - \mathsf{c}_{\mathsf{y}}) \log \frac{(\mathsf{s}_{\mathsf{h}} + \mathsf{s}_{\mathsf{y}})(1 - \mathsf{c}_{\mathsf{y}})}{\mathsf{s}_{\mathsf{y}}(2 - \mathsf{c}_{\mathsf{h}} - \mathsf{c}_{\mathsf{y}})} + (1 - \mathsf{c}_{\mathsf{h}}) \log \frac{(\mathsf{s}_{\mathsf{h}} + \mathsf{s}_{\mathsf{y}})(1 - \mathsf{c}_{\mathsf{h}})}{\mathsf{s}_{\mathsf{h}}(2 - \mathsf{c}_{\mathsf{h}} - \mathsf{c}_{\mathsf{y}})}$$

$$\text{(supremum is attained when } \mu^* = \tfrac{-(1-\mathsf{c}_{\mathsf{h}})\mathsf{s}_{\mathsf{h}} + (1-\mathsf{c}_{\mathsf{y}})\mathsf{s}_{\mathsf{y}}}{2 - \mathsf{c}_{\mathsf{y}} - \mathsf{c}_{\mathsf{h}}})$$

$$\ge (1 - \mathsf{c}_{\mathsf{y}}) \log \frac{2(1 - \mathsf{c}_{\mathsf{y}})}{(2 - \mathsf{c}_{\mathsf{h}} - \mathsf{c}_{\mathsf{y}})} + (1 - \mathsf{c}_{\mathsf{h}}) \log \frac{2(1 - \mathsf{c}_{\mathsf{h}})}{(2 - \mathsf{c}_{\mathsf{h}} - \mathsf{c}_{\mathsf{y}})}$$

$$\text{(minimum is attained when } \mathsf{s}_{\mathsf{h}} = \mathsf{s}_{\mathsf{y}} \text{ since } \mathsf{c}_{\mathsf{h}} \ge \mathsf{c}_{\mathsf{y}} \text{ and } \mathsf{s}_{\mathsf{h}} \ge \mathsf{s}_{\mathsf{y}})$$

$$\ge \frac{(\mathsf{c}_{\mathsf{h}} - \mathsf{c}_{\mathsf{y}})^2}{2(2 - \mathsf{c}_{\mathsf{h}} - \mathsf{c}_{\mathsf{y}})} \qquad\qquad (a \log \tfrac{2a}{a+b} + b \log \tfrac{2b}{a+b} \ge \tfrac{(a-b)^2}{2(a+b)}, \forall a, b \in [0, 1])$$

$$\ge \frac{(\mathsf{c}_{\mathsf{h}} - \mathsf{c}_{\mathsf{y}})^2}{4}.$$

Therefore, by Lemma B.2, $\Delta\mathcal{C}_{\mathsf{L}_{\mathrm{ham}}, \mathcal{H}}(h, x) \le 2\Big(\Delta\mathcal{C}_{\widetilde{\mathsf{L}}_{\mathrm{log}}, \mathcal{H}}(h, x)\Big)^{\frac{1}{2}}$. By Theorem B.1, we complete the proof. □

## B.3 Proof of Theorem 5.1

**Theorem 5.1.** *Let $\mathcal{H} = \mathcal{F}^l$. Assume that $\mathcal{F}$ is complete. Then, the following $\mathcal{H}$-consistency bound holds in the multi-label learning:*

$$\forall h \in \mathcal{H}, \quad \mathcal{R}_{\mathsf{L}}(h) - \mathcal{R}_{\mathsf{L}}^*(\mathcal{H}) + \mathcal{M}_{\mathsf{L}}(\mathcal{H}) \le 2\Big(\mathcal{R}_{\widetilde{\mathsf{L}}_{\log}}(h) - \mathcal{R}_{\widetilde{\mathsf{L}}_{\log}}^*(\mathcal{H}) + \mathcal{M}_{\widetilde{\mathsf{L}}_{\log}}(\mathcal{H})\Big)^{\frac{1}{2}}. \tag{18}$$

*Proof.* The proof is basically the same as that of Theorem 4.1, modulo replacing the Hamming loss $\mathsf{L}_{\mathrm{ham}}$ with a general multi-label loss $\mathsf{L}$. We adopt the following notation: $\mathsf{c}_\mathsf{h} = \mathbb{E}_{y|x}\big[\overline{\mathsf{L}}(\mathsf{h}, y)\big]$, $\mathsf{c}_\mathsf{y} = \mathbb{E}_{y|x}\big[\overline{\mathsf{L}}(\mathsf{y}, y)\big]$ and $\mathsf{c}_{y'} = \mathbb{E}_{y|x}\big[\overline{\mathsf{L}}(y', y)\big]$, $\forall y' \in \mathcal{Y}$. We will denote by $\mathsf{s}(h, x, y') = \frac{e^{\sum_{i=1}^{l} y_i' h(x,i)}}{\sum_{y'' \in \mathcal{Y}} e^{\sum_{i=1}^{l} y_i'' h(x,i)}}$ and simplify notation by using $\mathsf{s}_{y'}$, thereby dropping the dependency on $h$ and $x$. It is clear that $\mathsf{s}_{y'} \in [0, 1]$. Then, the conditional error of $\widetilde{\mathsf{L}}_{\log}$ can be expressed as follows:

$$\begin{aligned}
\mathcal{C}_{\widetilde{\mathsf{L}}_{\log}}(h, x) &= \mathbb{E}_{y|x}\left[ \sum_{y' \in \mathcal{Y}} \big(1 - \overline{\mathsf{L}}(y', y)\big) \log\left( \sum_{y'' \in \mathcal{Y}} e^{\sum_{i=1}^{l} (y_i'' - y_i') h(x,i)} \right) \right] \\
&= -\sum_{y' \in \mathcal{Y}} (1 - \mathsf{c}_{y'}) \log(\mathsf{s}_{y'})
\end{aligned}$$

For any $\mathsf{h} \ne \mathsf{y}$, we define $\mathsf{s}^\mu$ as follows: set $\mathsf{s}_{y'}^\mu = \mathsf{s}_{y'}$ for all $y' \ne \mathsf{y}$ and $y' \ne \mathsf{h}$; define $\mathsf{s}_\mathsf{h}^\mu = \mathsf{s}_\mathsf{y} - \mu$; and let $\mathsf{s}_\mathsf{y}^\mu = \mathsf{s}_\mathsf{h} + \mu$. Note that $\mathsf{s}^\mu$ can be realized by some $h' \in \mathcal{H}$ under the assumption. Then, we have

$$\begin{aligned}
\Delta\mathcal{C}_{\widetilde{\mathsf{L}}_{\log},\mathcal{H}}(h, x) &\ge \left( -\sum_{y' \in \mathcal{Y}} (1 - \mathsf{c}_{y'}) \log(\mathsf{s}_{y'}) \right) - \inf_{\mu \in \mathbb{R}}\left( -\sum_{y' \in \mathcal{Y}} (1 - \mathsf{c}_{y'}) \log(\mathsf{s}_{y'}^\mu) \right) \\
&= \sup_{\mu \in \mathbb{R}}\{(1 - \mathsf{c}_\mathsf{h})[\log(\mathsf{s}_\mathsf{y} - \mu) - \log(\mathsf{s}_\mathsf{h})] + (1 - \mathsf{c}_\mathsf{y})[\log(\mathsf{s}_\mathsf{h} + \mu) - \log(\mathsf{s}_\mathsf{y})]\} \\
&= (1 - \mathsf{c}_\mathsf{y}) \log \frac{(\mathsf{s}_\mathsf{h} + \mathsf{s}_\mathsf{y})(1 - \mathsf{c}_\mathsf{y})}{\mathsf{s}_\mathsf{y}(2 - \mathsf{c}_\mathsf{h} - \mathsf{c}_\mathsf{y})} + (1 - \mathsf{c}_\mathsf{h}) \log \frac{(\mathsf{s}_\mathsf{h} + \mathsf{s}_\mathsf{y})(1 - \mathsf{c}_\mathsf{h})}{\mathsf{s}_\mathsf{h}(2 - \mathsf{c}_\mathsf{h} - \mathsf{c}_\mathsf{y})} \\
&\qquad\qquad \text{(supremum is attained when } \mu^* = \tfrac{-(1-\mathsf{c}_\mathsf{h})\mathsf{s}_\mathsf{h} + (1-\mathsf{c}_\mathsf{y})\mathsf{s}_\mathsf{y}}{2 - \mathsf{c}_\mathsf{y} - \mathsf{c}_\mathsf{h}}) \\
&\ge (1 - \mathsf{c}_\mathsf{y}) \log \frac{2(1 - \mathsf{c}_\mathsf{y})}{(2 - \mathsf{c}_\mathsf{h} - \mathsf{c}_\mathsf{y})} + (1 - \mathsf{c}_\mathsf{h}) \log \frac{2(1 - \mathsf{c}_\mathsf{h})}{(2 - \mathsf{c}_\mathsf{h} - \mathsf{c}_\mathsf{y})} \\
&\qquad\qquad\qquad\qquad \text{(minimum is attained when } \mathsf{s}_\mathsf{h} = \mathsf{s}_\mathsf{y}) \\
&\ge \frac{(\mathsf{c}_\mathsf{h} - \mathsf{c}_\mathsf{y})^2}{2(2 - \mathsf{c}_\mathsf{h} - \mathsf{c}_\mathsf{y})} \qquad\qquad (a \log \tfrac{2a}{a+b} + b \log \tfrac{2b}{a+b} \ge \tfrac{(a-b)^2}{2(a+b)}, \forall a, b \in [0, 1]) \\
&\ge \frac{(\mathsf{c}_\mathsf{h} - \mathsf{c}_\mathsf{y})^2}{4}.
\end{aligned}$$

Therefore, by Lemma B.2, $\Delta\mathcal{C}_{\mathsf{L},\mathcal{H}}(h, x) \le 2\Big(\Delta\mathcal{C}_{\widetilde{\mathsf{L}}_{\log},\mathcal{H}}(h, x)\Big)^{\frac{1}{2}}$. By Theorem B.1, we complete the proof. $\qquad\square$

## B.4 Proof of Theorem 6.1

**Theorem 6.1.** *Let $\mathcal{H} = \mathcal{F}^l$. Assume that $\mathcal{F}$ is complete. Then, the following $\mathcal{H}$-consistency bound holds in the multi-label learning:*

$$\forall h \in \mathcal{H}, \quad \mathcal{R}_\mathsf{L}(h) - \mathcal{R}_\mathsf{L}^*(\mathcal{H}) + \mathcal{M}_\mathsf{L}(\mathcal{H}) \leq \Gamma\Big(\mathcal{R}_{\widetilde{\mathsf{L}}_\mathrm{comp}}(h) - \mathcal{R}_{\widetilde{\mathsf{L}}_\mathrm{comp}}^*(\mathcal{H}) + \mathcal{M}_{\widetilde{\mathsf{L}}_\mathrm{comp}}(\mathcal{H})\Big), \qquad (21)$$

*where $\Gamma(t) = 2\sqrt{t}$ when $\Psi(u) = \log(u)$ or $u - 1$; $\Gamma(t) = 2\sqrt{n^q t}$ when $\Psi(u) = \frac{1}{q}\big(1 - \frac{1}{u^q}\big), q \in (0,1)$; and $\Gamma(t) = nt$ when $\Psi(u) = 1 - \frac{1}{u}$.*

*Proof.* Recall that we adopt the following notation: $\mathsf{c}_\mathsf{h} = \mathbb{E}_{y|x}\big[\overline{\mathsf{L}}(\mathsf{h}, y)\big]$, $\mathsf{c}_\mathsf{y} = \mathbb{E}_{y|x}\big[\overline{\mathsf{L}}(\mathsf{y}, y)\big]$ and $\mathsf{c}_{y'} = \mathbb{E}_{y|x}\big[\overline{\mathsf{L}}(y', y)\big]$, $\forall y' \in \mathcal{Y}$. We will denote by $\mathsf{s}(h, x, y') = \frac{e^{\sum_{i=1}^l y_i' h(x,i)}}{\sum_{y'' \in \mathcal{Y}} e^{\sum_{i=1}^l y_i'' h(x,i)}}$ and simplify notation by using $\mathsf{s}_{y'}$, thereby dropping the dependency on $h$ and $x$. It is clear that $\mathsf{s}_{y'} \in [0,1]$. Next, we will analyze case by case.

**The case where** $\Phi(u) = \log(u)$: See the proof of Theorem 5.1.

**The case where** $\Phi(u) = u - 1$: The conditional error of $\widetilde{\mathsf{L}}_\mathrm{comp}$ can be expressed as follows:

$$\mathcal{C}_{\widetilde{\mathsf{L}}_\mathrm{comp}}(h, x)$$
$$= \mathbb{E}_{y|x}\left[\sum_{y' \in \mathcal{Y}}\big(1 - \overline{\mathsf{L}}(y', y)\big)\left(\sum_{y'' \in \mathcal{Y}} e^{\sum_{i=1}^l (y_i'' - y_i') h(x,i)} - 1\right)\right]$$
$$= \sum_{y' \in \mathcal{Y}}(1 - \mathsf{c}_{y'})\left(\frac{1}{\mathsf{s}_{y'}} - 1\right).$$

For any $\mathsf{h} \neq \mathsf{y}$, we define $\mathsf{s}^\mu$ as follows: set $\mathsf{s}_{y'}^\mu = \mathsf{s}_{y'}$ for all $y' \neq \mathsf{y}$ and $y' \neq \mathsf{h}$; define $\mathsf{s}_\mathsf{h}^\mu = \mathsf{s}_\mathsf{y} - \mu$; and let $\mathsf{s}_\mathsf{y}^\mu = \mathsf{s}_\mathsf{h} + \mu$. Note that $\mathsf{s}^\mu$ can be realized by some $h' \in \mathcal{H}$ under the assumption. Then, we have

$$\Delta\mathcal{C}_{\widetilde{\mathsf{L}}_\mathrm{comp}, \mathcal{H}}(h, x)$$
$$\geq \sum_{y' \in \mathcal{Y}}(1 - \mathsf{c}_{y'})\left(\frac{1}{\mathsf{s}_{y'}} - 1\right) - \inf_{\mu \in \mathbb{R}}\left(\sum_{y' \in \mathcal{Y}}(1 - \mathsf{c}_{y'})\left(\frac{1}{\mathsf{s}_{y'}^\mu} - 1\right)\right)$$
$$= \sup_{\mu \in \mathbb{R}}\left\{(1 - \mathsf{c}_\mathsf{h})\left[\frac{1}{\mathsf{s}_\mathsf{h}} - \frac{1}{\mathsf{s}_\mathsf{y} - \mu}\right] + (1 - \mathsf{c}_\mathsf{y})\left[\frac{1}{\mathsf{s}_\mathsf{y}} - \frac{1}{\mathsf{s}_\mathsf{h} + \mu}\right]\right\}$$
$$= \frac{1 - \mathsf{c}_\mathsf{h}}{\mathsf{s}_\mathsf{h}} + \frac{1 - \mathsf{c}_\mathsf{y}}{\mathsf{s}_\mathsf{y}} - \frac{2 - \mathsf{c}_\mathsf{h} - \mathsf{c}_\mathsf{y} + 2(1 - \mathsf{c}_\mathsf{h})^{\frac{1}{2}}(1 - \mathsf{c}_\mathsf{y})^{\frac{1}{2}}}{\mathsf{s}_\mathsf{h} + \mathsf{s}_\mathsf{y}}$$

$$\text{(supremum is attained when } \mu^* = \frac{-\sqrt{1 - \mathsf{c}_\mathsf{h}}\mathsf{s}_\mathsf{h} + \sqrt{1 - \mathsf{c}_\mathsf{y}}\mathsf{s}_\mathsf{y}}{\sqrt{1 - \mathsf{c}_\mathsf{y}} + \sqrt{1 - \mathsf{c}_\mathsf{h}}})$$

$$\geq \Big((1 - \mathsf{c}_\mathsf{h})^{\frac{1}{2}} - (1 - \mathsf{c}_\mathsf{y})^{\frac{1}{2}}\Big)^2 \qquad \text{(minimum is attained when } \mathsf{s}_\mathsf{h} = \mathsf{s}_\mathsf{y} = \tfrac{1}{2})$$

$$= \frac{(\mathsf{c}_\mathsf{h} - \mathsf{c}_\mathsf{y})^2}{\Big((1 - \mathsf{c}_\mathsf{h})^{\frac{1}{2}} + (1 - \mathsf{c}_\mathsf{y})^{\frac{1}{2}}\Big)^2}$$

$$\geq \frac{(\mathsf{c}_\mathsf{h} - \mathsf{c}_\mathsf{y})^2}{4}.$$

Therefore, by Lemma B.2, $\Delta\mathcal{C}_{\mathsf{L}, \mathcal{H}}(h, x) \leq 2\Big(\Delta\mathcal{C}_{\widetilde{\mathsf{L}}_\mathrm{comp}, \mathcal{H}}(h, x)\Big)^{\frac{1}{2}}$. By Theorem B.1, we complete the proof.

**The case where** $\Phi(u) = \frac{1}{q}\big(1 - \frac{1}{u^q}\big), q \in (0,1)$: The conditional error of $\widetilde{\mathsf{L}}_\mathrm{comp}$ can be expressed as:

$$\mathcal{C}_{\widetilde{\mathsf{L}}_\mathrm{comp}}(h, x) = \frac{1}{q}\sum_{y' \in \mathcal{Y}}(1 - \mathsf{c}_{y'})(1 - (\mathsf{s}_{y'})^q).$$

For any $h \neq y$, we define $s^\mu$ as follows: set $s^\mu_{y'} = s_{y'}$ for all $y' \neq y$ and $y' \neq h$; define $s^\mu_h = s_y - \mu$; and let $s^\mu_y = s_h + \mu$. Note that $s^\mu$ can be realized by some $h' \in \mathcal{H}$ under the assumption. Then, we have

$$\Delta\mathcal{C}_{\widetilde{\mathsf{L}}_{\mathrm{comp}},\mathcal{H}}(h,x)$$

$$\geq \frac{1}{q}\sum_{y'\in\mathcal{Y}}(1-c_{y'})(1-s_y) - \inf_{\mu\in\mathbb{R}}\left(\frac{1}{q}\sum_{y'\in\mathcal{Y}}(1-c_{y'})\left(1-(s^\mu_{y'})^q\right)\right)$$

$$= \frac{1}{q}\sup_{\mu\in\mathbb{R}}\{(1-c_h)[-s_h + (s_y-\mu)^q] + (1-c_y)[-(s_y)^q + (s_h+\mu)^q]\}$$

$$= \frac{1}{q}(s_h+s_y)^q\left((1-c_y)^{\frac{1}{1-q}} + (1-c_h)^{\frac{1}{1-q}}\right)^{1-q} - \frac{1}{q}(1-c_y)s_y^q - \frac{1}{q}(1-c_h)s_h^q$$

$$\text{(supremum is attained when } \mu^* = \tfrac{-(1-c_h)^{\frac{1}{1-q}}s_h+(1-c_y)^{\frac{1}{1-q}}s_y}{(1-c_y)^{\frac{1}{1-q}}+(1-c_h)^{\frac{1}{1-q}}})$$

$$\geq \frac{1}{qn^q}\left[2^q\left((1-c_y)^{\frac{1}{1-q}} + (1-c_h)^{\frac{1}{1-q}}\right)^{1-q} - (1-c_y) - (1-c_h)\right]$$

$$\text{(minimum is attained when } s_h = s_y = \tfrac{1}{n})$$

$$\geq \frac{(c_h-c_y)^2}{4n^q}. \qquad \left(\left(\tfrac{a^{\frac{1}{1-q}}+b^{\frac{1}{1-q}}}{2}\right)^{1-q} - \tfrac{a+b}{2} \geq \tfrac{q}{4}(a-b)^2, \forall a,b\in[0,1], 0\leq a+b\leq 1\right)$$

Therefore, by Lemma B.2, $\Delta\mathcal{C}_{\mathsf{L},\mathcal{H}}(h,x) \leq 2n^{\frac{q}{2}}\left(\Delta\mathcal{C}_{\widetilde{\mathsf{L}}_{\mathrm{comp}},\mathcal{H}}(h,x)\right)^{\frac{1}{2}}$. By Theorem B.1, we complete the proof.

**The case where** $\Phi(u) = \left(1 - \frac{1}{u}\right)$: The conditional error of $\widetilde{\mathsf{L}}_{\mathrm{comp}}$ can be expressed as:

$$\mathcal{C}_{\widetilde{\mathsf{L}}_{\mathrm{comp}}}(h,x) = \sum_{y'\in\mathcal{Y}}(1-c_{y'})(1-(s_{y'})^q).$$

For any $h \neq y$, we define $s^\mu$ as follows: set $s^\mu_{y'} = s_{y'}$ for all $y' \neq y$ and $y' \neq h$; define $s^\mu_h = s_y - \mu$; and let $s^\mu_y = s_h + \mu$. Note that $s^\mu$ can be realized by some $h' \in \mathcal{H}$ under the assumption. Then, we have

$$\Delta\mathcal{C}_{\widetilde{\mathsf{L}}_{\mathrm{comp}},\mathcal{H}}(h,x)$$

$$\geq \sum_{y'\in\mathcal{Y}}(1-c_{y'})(1-s_y) - \inf_{\mu\in\mathbb{R}}\left(\sum_{y'\in\mathcal{Y}}(1-c_{y'})\left(1-s^\mu_{y'}\right)\right)$$

$$= \sup_{\mu\in\mathbb{R}}\{(1-c_h)[-s_h + s_y - \mu] + (1-c_y)[-s_y + s_h + \mu]\}$$

$$= s_h(c_h - c_y) \qquad\qquad \text{(supremum is attained when } \mu^* = s_y)$$

$$\geq \frac{1}{n}(c_h - c_y). \qquad\qquad \text{(minimum is attained when } s_h = \tfrac{1}{n})$$

Therefore, by Lemma B.2, $\Delta\mathcal{C}_{\mathsf{L},\mathcal{H}}(h,x) \leq n\Delta\mathcal{C}_{\widetilde{\mathsf{L}}_{\mathrm{comp}},\mathcal{H}}(h,x)$. By Theorem B.1, we complete the proof. $\qquad\square$

## B.5  Proof of Theorem 7.1

**Theorem 7.1.** *Let $\mathcal{H} = \mathcal{F}^l$. Assume that $\mathcal{F}$ is complete Then, the following $\mathcal{H}$-consistency bound holds in the multi-label learning:*

$$\forall h \in \mathcal{H}, \quad \mathcal{R}_{\mathsf{L}}(h) - \mathcal{R}_{\mathsf{L}}^*(\mathcal{H}) + \mathcal{M}_{\mathsf{L}}(\mathcal{H}) \leq \Gamma\Big(\mathcal{R}_{\widetilde{\mathsf{L}}_{\mathrm{cstnd}}}(h) - \mathcal{R}_{\widetilde{\mathsf{L}}_{\mathrm{cstnd}}}^*(\mathcal{H}) + \mathcal{M}_{\widetilde{\mathsf{L}}_{\mathrm{cstnd}}}(\mathcal{H})\Big), \qquad (24)$$

*where $\Gamma(t) = 2\sqrt{\mathsf{L}_{\max}t}$ when $\Phi(u) = e^{-u}$; $\Gamma(t) = 2\sqrt{t}$ when $\Phi(u) = \max\{0, 1-u\}^2$; and $\Gamma(t) = t$ when $\Phi(u) = \max\{0, 1-u\}$ or $\Phi(u) = \min\{\max\{0, 1 - u/\rho\}, 1\}$, $\rho > 0$.*

*Proof.* Recall that we adopt the following notation: $\mathsf{c}_{\mathsf{h}} = \mathbb{E}_{y|x}\big[\overline{\mathsf{L}}(\mathsf{h}, y)\big]$, $\mathsf{c}_{\mathsf{y}} = \mathbb{E}_{y|x}\big[\overline{\mathsf{L}}(\mathsf{y}, y)\big]$ and $\mathsf{c}_{y'} = \mathbb{E}_{y|x}\big[\overline{\mathsf{L}}(y', y)\big]$, $\forall y' \in \mathcal{Y}$. We will also denote by $\mathsf{z}(h, x, y') = \sum_{i=1}^l y_i' h(x, i)$ and simplify notation by using $\mathsf{z}_{y'}$, thereby dropping the dependency on $h$ and $x$. It is clear that the constraint can be expressed as $\sum_{y' \in \mathcal{Y}} \mathsf{z}_{y'} = 0$. Next, we will analyze case by case.

**The case where $\Phi(u) = e^{-u}$:** The conditional error of $\widetilde{\mathsf{L}}_{\mathrm{cstnd}}$ can be expressed as follows:

$$\mathcal{C}_{\widetilde{\mathsf{L}}_{\mathrm{cstnd}}}(h, x) = \mathbb{E}_{y|x}\left[\sum_{y' \in \mathcal{Y}} \overline{\mathsf{L}}(y', y) e^{\sum_{i=1}^l y_i' h(x, i)}\right] = \sum_{y' \in \mathcal{Y}} \mathsf{c}_{y'} e^{\mathsf{z}_{y'}}.$$

For any $\mathsf{h} \neq \mathsf{y}$, we define $\mathsf{z}^\mu$ as follows: set $\mathsf{z}_{y'}^\mu = \mathsf{z}_{y'}$ for all $y' \neq \mathsf{y}$ and $y' \neq \mathsf{h}$; define $\mathsf{z}_{\mathsf{h}}^\mu = \mathsf{z}_{\mathsf{y}} - \mu$; and let $\mathsf{z}_{\mathsf{y}}^\mu = \mathsf{z}_{\mathsf{h}} + \mu$. Note that $\mathsf{z}^\mu$ can be realized by some $h' \in \mathcal{H}$ under the assumption. Then, we have

$$\begin{aligned}
\Delta\mathcal{C}_{\widetilde{\mathsf{L}}_{\mathrm{comp}}, \mathcal{H}}(h, x) &\geq \sum_{y' \in \mathcal{Y}} \mathsf{c}_{y'} e^{\mathsf{z}_{y'}} - \inf_{\mu \in \mathbb{R}}\left(\sum_{y' \in \mathcal{Y}} \mathsf{c}_{y'} e^{\mathsf{z}_{y'}^\mu}\right)\\
&= \sup_{\mu \in \mathbb{R}}\left\{\mathsf{c}_{\mathsf{y}}\big(e^{\mathsf{z}_{\mathsf{y}}} - e^{\mathsf{z}_{\mathsf{h}} + \mu}\big) + \mathsf{c}_{\mathsf{h}}\big(e^{\mathsf{z}_{\mathsf{h}}} - e^{\mathsf{z}_{\mathsf{y}} - \mu}\big)\right\}\\
&= \left(\sqrt{\mathsf{c}_{\mathsf{h}} e^{\mathsf{z}_{\mathsf{h}}}} - \sqrt{\mathsf{c}_{\mathsf{y}} e^{\mathsf{z}_{\mathsf{y}}}}\right)^2 \qquad \text{(supremum is attained when } \mu^* = \tfrac{1}{2}\log\tfrac{\mathsf{c}_{\mathsf{y}} e^{\mathsf{z}_{\mathsf{y}}}}{\mathsf{c}_{\mathsf{h}} e^{\mathsf{z}_{\mathsf{h}}}}\text{)}\\
&= \left(\frac{\mathsf{c}_{\mathsf{h}} - \mathsf{c}_{\mathsf{y}}}{\sqrt{\mathsf{c}_{\mathsf{y}}} + \sqrt{\mathsf{c}_{\mathsf{h}}}}\right)^2 \qquad\qquad \text{(minimum is attained when } \mathsf{z}_{\mathsf{h}} = \mathsf{z}_{\mathsf{y}} = 0\text{)}\\
&\geq \frac{1}{4\mathsf{L}_{\max}}\big(\mathsf{c}_{\mathsf{h}} - \mathsf{c}_{\mathsf{y}}\big)^2.
\end{aligned}$$

Therefore, by Lemma B.2, $\Delta\mathcal{C}_{\mathsf{L}, \mathcal{H}}(h, x) \leq 2(\mathsf{L}_{\max})^{\frac{1}{2}}\big(\Delta\mathcal{C}_{\widetilde{\mathsf{L}}_{\mathrm{cstnd}}, \mathcal{H}}(h, x)\big)^{\frac{1}{2}}$. By Theorem B.1, we complete the proof.

**The case where $\Phi(u) = \max\{0, 1-u\}^2$:** The conditional error of $\widetilde{\mathsf{L}}_{\mathrm{cstnd}}$ can be expressed as follows:

$$\mathcal{C}_{\widetilde{\mathsf{L}}_{\mathrm{cstnd}}}(h, x) = \sum_{y' \in \mathcal{Y}} \mathsf{c}_{y'} \max\{0, 1 + \mathsf{z}_{y'}\}^2.$$

For any $\mathsf{h} \neq \mathsf{y}$, we define $\mathsf{z}^\mu$ as follows: set $\mathsf{z}_{y'}^\mu = \mathsf{z}_{y'}$ for all $y' \neq \mathsf{y}$ and $y' \neq \mathsf{h}$; define $\mathsf{z}_{\mathsf{h}}^\mu = \mathsf{z}_{\mathsf{y}} - \mu$; and let $\mathsf{z}_{\mathsf{y}}^\mu = \mathsf{z}_{\mathsf{h}} + \mu$. Note that $\mathsf{z}^\mu$ can be realized by some $h' \in \mathcal{H}$ under the assumption. Then, we have

$$\begin{aligned}
&\Delta\mathcal{C}_{\widetilde{\mathsf{L}}_{\mathrm{cstnd}}, \mathcal{H}}(h, x)\\
&\geq \sum_{y' \in \mathcal{Y}} \mathsf{c}_{y'} \max\{0, 1 + \mathsf{z}_{y'}\}^2 - \inf_{\mu \in \mathbb{R}}\left(\sum_{y' \in \mathcal{Y}} \mathsf{c}_{y'} \max\{0, 1 + \mathsf{z}_{y'}^\mu\}^2\right)\\
&= \sup_{\mu \in \mathbb{R}}\left\{\mathsf{c}_{\mathsf{y}}\Big(\max\{0, 1 + \mathsf{z}_{\mathsf{y}}\}^2 - \max\{0, 1 + \mathsf{z}_{\mathsf{h}} + \mu\}^2\Big) + \mathsf{c}_{\mathsf{h}}\Big(\max\{0, 1 + \mathsf{z}_{\mathsf{h}}\}^2 - \max\{0, 1 + \mathsf{z}_{\mathsf{y}} - \mu\}^2\Big)\right\}\\
&\geq \big(1 + \mathsf{z}_{\mathsf{h}}\big)^2 \big(\mathsf{c}_{\mathsf{y}} - \mathsf{c}_{\mathsf{h}}\big)^2 \qquad\qquad \text{(differentiating with respect to } \mu \text{ to optimize)}\\
&\geq \big(\mathsf{c}_{\mathsf{h}} - \mathsf{c}_{\mathsf{y}}\big)^2. \qquad\qquad\qquad \text{(minimum is attained when } \mathsf{z}_{\mathsf{h}} = 0\text{)}
\end{aligned}$$

Therefore, by Lemma B.2, $\Delta\mathcal{C}_{\mathsf{L}, \mathcal{H}}(h, x) \leq \big(\Delta\mathcal{C}_{\widetilde{\mathsf{L}}_{\mathrm{cstnd}}, \mathcal{H}}(h, x)\big)^{\frac{1}{2}}$. By Theorem B.1, we complete the proof.

**The case where** $\Phi(u) = \max\{0, 1 - u\}$: The conditional error of $\widetilde{\mathsf{L}}_{\mathrm{cstnd}}$ can be expressed as:

$$\mathcal{C}_{\widetilde{\mathsf{L}}_{\mathrm{cstnd}}}(h, x) = \sum_{y' \in \mathcal{Y}} \mathsf{c}_{y'} \max\{0, 1 + \mathsf{z}_{y'}\}.$$

For any $h \neq y$, we define $\mathsf{z}^\mu$ as follows: set $\mathsf{z}_{y'}^\mu = \mathsf{z}_{y'}$ for all $y' \neq y$ and $y' \neq h$; define $\mathsf{z}_h^\mu = \mathsf{z}_y - \mu$; and let $\mathsf{z}_y^\mu = \mathsf{z}_h + \mu$. Note that $\mathsf{z}^\mu$ can be realized by some $h' \in \mathcal{H}$ under the assumption. Then, we have

$$\Delta\mathcal{C}_{\widetilde{\mathsf{L}}_{\mathrm{cstnd}}, \mathcal{H}}(h, x)$$

$$\geq \sum_{y' \in \mathcal{Y}} \mathsf{c}_{y'} \max\{0, 1 + \mathsf{z}_{y'}\} - \inf_{\mu \in \mathbb{R}} \left( \sum_{y' \in \mathcal{Y}} \mathsf{c}_{y'} \max\{0, 1 + \mathsf{z}_{y'}^\mu\} \right)$$

$$= \sup_{\mu \in \mathbb{R}} \left\{ \mathsf{c}_y \big( \max\{0, 1 + \mathsf{z}_y\} - \max\{0, 1 + \mathsf{z}_h + \mu\} \big) + \mathsf{c}_h \big( \max\{0, 1 + \mathsf{z}_h\}^2 - \max\{0, 1 + \mathsf{z}_y - \mu\}^2 \big) \right\}$$

$$\geq (1 + \mathsf{z}_h)(\mathsf{c}_y - \mathsf{c}_h) \qquad\qquad \text{(differentiating with respect to } \mu \text{ to optimize)}$$

$$\geq (\mathsf{c}_h - \mathsf{c}_y). \qquad\qquad\qquad \text{(minimum is attained when } \mathsf{z}_h = 0)$$

Therefore, by Lemma B.2, $\Delta\mathcal{C}_{\mathsf{L}, \mathcal{H}}(h, x) \leq \Delta\mathcal{C}_{\widetilde{\mathsf{L}}_{\mathrm{cstnd}}, \mathcal{H}}(h, x)$. By Theorem B.1, we complete the proof.

**The case where** $\Phi(u) = \min\{\max\{0, 1 - u/\rho\}, 1\}, \rho > 0$: The conditional error of $\widetilde{\mathsf{L}}_{\mathrm{cstnd}}$ can be expressed as:

$$\mathcal{C}_{\widetilde{\mathsf{L}}_{\mathrm{cstnd}}}(h, x) = \sum_{y' \in \mathcal{Y}} \mathsf{c}_{y'} \min\{\max\{0, 1 + \mathsf{z}_{y'}/\rho\}, 1\}.$$

For any $h \neq y$, we define $\mathsf{z}^\mu$ as follows: set $\mathsf{z}_{y'}^\mu = \mathsf{z}_{y'}$ for all $y' \neq y$ and $y' \neq h$; define $\mathsf{z}_h^\mu = \mathsf{z}_y - \mu$; and let $\mathsf{z}_y^\mu = \mathsf{z}_h + \mu$. Note that $\mathsf{z}^\mu$ can be realized by some $h' \in \mathcal{H}$ under the assumption. Then, we have

$$\Delta\mathcal{C}_{\widetilde{\mathsf{L}}_{\mathrm{cstnd}}, \mathcal{H}}(h, x)$$

$$\geq \sum_{y' \in \mathcal{Y}} \mathsf{c}_{y'} \min\{\max\{0, 1 + \mathsf{z}_{y'}/\rho\}, 1\} - \inf_{\mu \in \mathbb{R}} \left( \sum_{y' \in \mathcal{Y}} \mathsf{c}_{y'} \min\{\max\{0, 1 + \mathsf{z}_{y'}^\mu/\rho\}, 1\} \right)$$

$$= \sup_{\mu \in \mathbb{R}} \left\{ \mathsf{c}_y \big( \min\{\max\{0, 1 + \mathsf{z}_y/\rho\}, 1\} - \min\{\max\{0, 1 + (\mathsf{z}_h + \mu)/\rho\}, 1\} \big) \right.$$

$$\left. + \mathsf{c}_h \big( \min\{\max\{0, 1 + \mathsf{z}_h/\rho\}, 1\} - \min\{\max\{0, 1 + (\mathsf{z}_y - \mu)/\rho\}\}, 1) \big) \right\}$$

$$\geq (\mathsf{c}_y - \mathsf{c}_h). \qquad\qquad \text{(differentiating with respect to } \mu \text{ to optimize)}$$

Therefore, by Lemma B.2, $\Delta\mathcal{C}_{\mathsf{L}, \mathcal{H}}(h, x) \leq \Delta\mathcal{C}_{\widetilde{\mathsf{L}}_{\mathrm{cstnd}}, \mathcal{H}}(h, x)$. By Theorem B.1, we complete the proof. $\qquad\square$

## C  Future work

While our work introduces a unified surrogate loss framework that is Bayes-consistent across any multi-label loss, thereby broadening the scope beyond previous approaches that established consistency only for particular loss functions, there remains an exciting opportunity for empirical comparison with surrogate losses tailored to specific loss functions—a direction we leave for future work. Furthermore, refining surrogate losses to theoretically enhance performance for specific target losses presents another promising avenue for research.

