# OpenReview forum: "Multi-Label Learning with Stronger Consistency Guarantees"
_NeurIPS.cc/2024/Conference — NeurIPS 2024 poster_

### Official Review · Reviewer_73SN · 2024-07-12

**Soundness:** 3
**Presentation:** 2
**Contribution:** 2
**Rating:** 5
**Confidence:** 1

**Summary:**

This paper proposes an improved approach to multi-label learning using $\mathcal{H}$-consistency bounds by introducing the multi-label logistic loss to effectively handle label correlations. It extends to various multi-label losses, ensuring Bayes-consistency across diverse settings, and includes efficient gradient computation algorithms for minimizing the proposed loss function. This work offers a unified framework with robust consistency guarantees, advancing beyond traditional methods in multi-label learning.

**Strengths:**

- Introducing the multi-label logistic loss, which effectively addresses label correlations often overlooked by traditional binary relevance surrogates under Hamming loss.

- The paper establishes $\mathcal{H}$-consistency bounds for a wide range of multi-label losses, ensuring Bayes-consistency across diverse multi-label learning scenarios. This extends beyond previous research that primarily focused on specific loss functions.

- It offers a unified framework that accommodates various multi-label losses, including novel extensions and adaptations from standard classification. This is supported by efficient gradient computation algorithms specifically designed for minimizing the proposed multi-label logistic loss.

**Weaknesses:**

- The motivation and background of this paper lack clear logic and hierarchy. It is suggested to first outline the shortcomings of existing methods and then clearly present the research questions addressed in this paper.

**Questions:**

Please check the weaknesses.

---

> ### Author Rebuttal · Authors · 2024-08-06
>
> Thank you for your encouraging review. We will take your suggestions into account when preparing the final version. Below please find responses to specific questions.
>
> **Weaknesses: The motivation and background of this paper lack clear logic and hierarchy. It is suggested to first outline the shortcomings of existing methods and then clearly present the research questions addressed in this paper.**
>
> **Response:** Thank you for the suggestion. Here is a list of shortcomings of existing methods and the research questions addressed in the paper:
>
> - Lack of Theoretical Analysis: Only a few studies focus on the theoretical analysis of multi-label learning, particularly the Bayes-consistency of surrogate losses. Can we present a comprehensive analysis of surrogate losses for multi-label learning and establish strong consistency guarantees?
>
> - Limited Bayes-Consistency: Existing methods only establish Bayes-consistency for specific loss functions. Can we derive a unified surrogate loss framework that is Bayes-consistent for any multi-label loss?
>
> - Drawback of Bayes-Consistency: Bayes-consistency is an asymptotic guarantee and does not provide convergence guarantees. It also applies only to the family of all measurable functions, unlike the restricted hypothesis sets typically used in practice. Can we leverage state-of-the-art consistency guarantees—$H$-consistency bounds—when designing surrogate loss functions for multi-label learning?
>
> - Sub-optimal Dependency on the Number of Labels: For the simplest form of multi-label loss, the popular Hamming loss, the well-known consistent binary relevance surrogate, when using smooth losses such as logistic losses, suffers from a sub-optimal dependency on the number of labels in terms of $H$-consistency bounds. Can we design smooth loss functions with improved dependency on the number of labels in their $H$-consistency bounds?
>
> - Label Correlations: One of the main concerns in multi-label learning is label correlations. The Bayes-consistent binary relevance surrogate fails to leverage label correlations. Can we design consistent loss functions that effectively benefit from label correlations as well?
>
> To address these drawbacks, we introduce a novel surrogate loss, multi-label logistic loss, that accounts for label correlations and benefits from label-independent $H$-consistency bounds. We then broaden our analysis to cover a more extensive family of multi-label losses, including all common ones and a new extension defined based on linear-fractional functions with respect to the confusion matrix.
>
> We also extend our multi-label logistic losses to more comprehensive multi-label comp-sum losses, adapting comp-sum losses from standard classification to multi-label learning. We prove that this family of surrogate losses benefits from $H$-consistency bounds, and thus Bayes-consistency, across any general multi-label loss. Our work thus proposes a unified surrogate loss framework that is Bayes-consistent for any multi-label loss, significantly expanding upon previous work which only established consistency for specific loss functions.
>
> Additionally, we adapt constrained losses from standard classification to multi-label constrained losses in a similar way, which also benefit from $H$-consistency bounds and thus Bayes-consistency for any multi-label loss. We further describe efficient gradient computation algorithms for minimizing the multi-label logistic loss. This unified framework holds promise for broader applications and opens new avenues for future research in multi-label learning and related areas.

---

> > ### Comment · Reviewer_73SN · 2024-08-12
> >
> > Thank you for thoroughly addressing my question and clarifying my doubts. As I am not familiar with this field, I will keep my score for now.

---

> > > ### Author Response · Authors · 2024-08-13
> > >
> > > Please let us know if we can provide further clarification regarding any question.

---

### Official Review · Reviewer_sTKD · 2024-07-12

**Soundness:** 3
**Presentation:** 3
**Contribution:** 3
**Rating:** 7
**Confidence:** 4

**Summary:**

The paper explores surrogate losses and algorithms for multi-label learning, focusing on \( \mathcal{H} \)-consistency bounds. It identifies the limitations of Hamming loss and introduces a new multi-label logistic loss that accounts for label correlations. The study extends this to a broader family of multi-label losses and adapts comp-sum losses from standard classification to multi-label learning. The authors propose a unified framework providing strong consistency guarantees for multi-label losses and describe efficient gradient computation methods for minimizing these losses.

**Strengths:**

1. The authors conduct a detailed analysis of the popular Hamming loss in multi-label learning when using smooth losses. They identify its sub-optimal dependency on the number of labels and its failure to account for label correlations, providing valuable insights into the limitations of existing loss functions.
1. The authors introduce an improvement by presenting a novel surrogate loss, the multi-label logistic loss, which accounts for label correlations and benefits from label-independent \( \mathcal{H} \)-consistency bounds. This innovation addresses the identified drawbacks of existing loss functions and broadens the analysis to include a more extensive family of multi-label losses, including a new extension based on linear-fractional functions related to the confusion matrix.
1. The authors extend their work by adapting multi-label logistic losses to more comprehensive multi-label comp-sum losses. By demonstrating that this family of surrogate losses benefits from \( \mathcal{H} \)-consistency bounds and Bayes-consistency across any general multi-label loss, they propose a unified surrogate loss framework. This expands upon previous work that only established consistency for specific loss functions, showcasing the applicability of their approach.
1. The authors' writing is clear and well-structured, with each theoretical assumption and conclusion articulated distinctly.

**Weaknesses:**

1. In section 4, although the excellent properties of the proposed multi-label logistic loss are proven, providing a detailed explanation of each component of this loss would further enhance the reader's understanding of its superiority.
2. If the advantages of this loss could be demonstrated through experimental validation, it would be more intuitive for readers.

**Questions:**

1. Could the authors elaborate on the individual components of the multi-label logistic loss and how each contributes to its overall effectiveness?
2. Given the detailed nature of this loss function, what is the computational complexity associated with implementing the multi-label logistic loss compared to other traditional loss functions? It is better to be able to verify the advantages and complexity of the algorithm through experiments.

**Limitations:**

The authors have adequately addressed the limitations.

---

> ### Author Rebuttal · Authors · 2024-08-06
>
> Thank you for your appreciation of our work. We will take your suggestions into account when preparing the final version. Below please find responses to specific questions.
>
> **Weakness 1. In section 4, although the excellent properties of the proposed multi-label logistic loss are proven, providing a detailed explanation of each component of this loss would further enhance the reader's understanding of its superiority.**
>
> **Question 1. Could the authors elaborate on the individual components of the multi-label logistic loss and how each contributes to its overall effectiveness?**
>
> **Response:** That's an excellent question. The component $\left( 1 - \overline{\mathsf{L}}_ {\mathrm{ham}}(\cdot, y) \right)$ acts as a weight vector for each logistic loss corresponding to the label $y'$. The term $\sum_{i = 1}^{l} \left( y''_ i - y'_ i \right) h(x, i)$ represents the difference in the scores between the label $y'$ and any other label $y''$, where these scores account for the correlations among the labels $y_i$ within the logarithmic function. The logarithmic term increases as the difference in scores increases. Therefore, the loss function imposes a greater penalty on larger differences through the penalty term $\left( 1 - \overline{\mathsf{L}}_ {\mathrm{ham}}(y', y) \right)$, which is dependent on the Hamming losses assigned to the prediction $y'$ and true label $y$. We will add a more detailed explanation in the final version.
>
> **Weakness  2. If the advantages of this loss could be demonstrated through experimental validation, it would be more intuitive for readers.**
>
> **Question 2. Given the detailed nature of this loss function, what is the computational complexity associated with implementing the multi-label logistic loss compared to other traditional loss functions? It is better to be able to verify the advantages and complexity of the algorithm through experiments.**
>
> **Response:** Thank you for your valuable feedback. As shown in Section 7, the computational complexity for optimizing and implementing the multi-label logistic loss is $O(l)$, modulo the precomputed quantities, which is comparable to that of other common multi-label surrogate losses.
>
> As you noted, this paper is primarily theoretical and algorithmic, and establishes a sound foundation for multi-label surrogate losses, backed by $H$-consistency bounds. Our framework offers a unique, unifying approach that ensures Bayes-consistency for any multi-label loss, a significant advantage over existing methods.
>
> While we have demonstrated that efficient algorithms can minimize multi-label logistic loss, we recognize the importance of further exploration. We agree that empirical comparisons with common multi-label surrogate losses would strengthen our work, and we will strive to include these in the final version.
>
> In future work, we are excited to expand upon this foundation with extensive empirical analyses.

---

> > ### Comment · Reviewer_sTKD · 2024-08-11
> >
> > Thank you for your reply. After referring to the comments of other reviewers, I decided to maintain my score.

---

> > > ### Author Response · Authors · 2024-08-13
> > >
> > > Thank you for your comments. We appreciate the reviewer's valuable feedback and constructive suggestions.

---

### Official Review · Reviewer_Abmm · 2024-07-12

**Soundness:** 3
**Presentation:** 3
**Contribution:** 3
**Rating:** 8
**Confidence:** 4

**Summary:**

The authors study surrogate losses and algorithms for multi-label learning via H-consistency bounds and introduce a novel surrogate loss, multi-label logistic loss in this paper. By broadening the H-consistency bounds analyses to more general multi-label losses and extending to multi-label comp-sum losses, the authors provide a unified surrogate loss framework for H-consistency.

**Strengths:**

1. This paper is well-written and easy to follow.
2. The authors make comprehensive reviews of related works, including their pros and cons.
3. The authors provide rigorous theoretical analyses of the limitations of existing binary relevance loss, the H-consistency of the proposed multi-label logistic loss, and the extensions to more general multi-label losses. The theoretical contribution is important for multi-label learning.
4. The authors demonstrate the efficient computation of the gradient for the proposed multi-label logistic loss and conduct time complexity analyses.

**Weaknesses:**

1. I understand that this is a theoretical work, and experiments of empirical evaluations are not its focus. However, adding experiments to compare the proposed loss with commonly used multi-label losses on standard datasets would make the paper more comprehensive and appealing. Besides, it can also verify whether the proposed loss is effective in practice.
2. There is a typo in line 300.($1-\bar{L}_{ham}(\cdot, y)$).

**Questions:**

See above weaknesses.

**Limitations:**

Yes, the authors have adequately addressed the limitations.

---

> ### Author Rebuttal · Authors · 2024-08-06
>
> Thank you for your appreciation of our work. We will take your suggestions into account when preparing the final version. Below please find responses to specific questions.
>
> **1. I understand that this is a theoretical work, and experiments of empirical evaluations are not its focus. However, adding experiments to compare the proposed loss with commonly used multi-label losses on standard datasets would make the paper more comprehensive and appealing. Besides, it can also verify whether the proposed loss is effective in practice.**
>
> **Response:** Thank you for your valuable feedback.
>
> As you noted, this paper is primarily theoretical and algorithmic, and establishes a sound foundation for multi-label surrogate losses, backed by $H$-consistency bounds.  Our framework offers a unique, unifying approach that ensures Bayes-consistency for any multi-label loss, a significant advantage over existing methods.
>
> While we have demonstrated that efficient algorithms can minimize multi-label logistic loss, we recognize the importance of further exploration. We agree that empirical comparisons with common multi-label surrogate losses would strengthen our work, and we will strive to include these in the final version.
>
> In future work, we are excited to expand upon this foundation with extensive empirical analyses.
>
> **2. There is a typo in line 300. $(1 - \overline{L}_{\mathrm{ham}}(\cdot, y)).$**
>
> **Response:** Thank you, we will correct it.

---

> ### Comment · Reviewer_Abmm · 2024-08-12
>
> Thanks for your responses. I have thoroughly reviewed the comments from other reviewers and the corresponding responses. I look forward to seeing the experimental results in the final version. I have no further questions at this time and have decided to maintain my current score.

---

> > ### Author Response · Authors · 2024-08-13
> >
> > Thank you for your comments. We will strive to include experimental results in the final version. We appreciate the reviewer’s support of our work and their valuable suggestions.

---

### Official Review · Reviewer_9NfQ · 2024-07-17

**Soundness:** 3
**Presentation:** 3
**Contribution:** 3
**Rating:** 5
**Confidence:** 2

**Summary:**

The paper derives H-consistency bounds for binary-relevance style surrogate losses, as well as a new surrogate, for mutli-label learning problems, showing that the proposed multi-label logistic loss whose upper-bound on the Hamming loss is independent of the number of labels.

**Strengths:**

The $H$-consistency bounds provided in the paper are more informative than existing Bayes-consistency results, as they hold not just in the infinite limit.

The novel multi-label logistic loss allows upper-bounds that do not depend on the number of labels.

**Weaknesses:**

The paper does not provide any experiments. While this is OK for a theory paper, it does mean that the question of whether the new surrogate works better in practice remains unanswered (which should be reflected in the conclusion section, at least), for two reasons:
a) all the theory provides are upper-bounds, which might not be indicative of actual performance
b) while the theory provides better guarantees for the task loss if the surrogate is reduced to the level $\epsilon$, it might be that reducing the new surrogate is just much more difficult than optimizing binary relevance. In particular, if the computational cost for reducing the multi-label logistic loss to the same level $\epsilon$ as binary relevance is larger by at least $\sqrt{l}$, then, normalized for compute, the advantage of the new surrogate vanishes.

It is claimed that the gradient of the multi-label logistic loss can be computed efficiently, yet the presented formulas still contain sums over the entire $2^l$ entries of the label space. Even if they can be precomputed once, already at moderate label space sizes of l ~ 100 would these quantities be intractable.

It is annoying that most equations are unnumbered. Even if they are not referred to in the paper, your readers and reviewers might want to reference them.

the equation after l. 328 switches between $\mathbf{\mathsf{y}}'$ and $y'$; and $y''$ changes to $y$

l. 114: I'm not sure what the point here is of introducing the threshold $t$, if it is set to $0$ in the same sentence? Couldn't $t$ be simply absorbed into $h$?

l. 178-180; 208: Arguably, completeness does _not_ hold in practice, because there is some form of upper-bound (e.g., weights representable in the given floating-point format)

l. 231. Binary relevance is not just Bayes-consistent w.r.t. the Hamming-loss, but also works for precision-at-$k$

In the equation after line 542, I think $\bar{L}$ should be $\bar{L}_\mathrm{ham}$?

l. 503: I think $q$ should be $q_i$, and there is a weird subscript on that line.

l. 174 consist -> consisting

**Questions:**

In several places, the paper talks about label correlations, in particular, it claims an advantage of the new surrogate is that it takes into account label correlations. However, it is never specified what exactly that means (conditional correlations, i.e., dependent on the specific instance $x$, or marginal correlations). Further, for many loss functions (such as Hamming-loss), the Bayes-optimal prediction is a function of purely the label marginals $P[Y_i|X]$, so it is not clear to me whether taking into account label correlations actually is an advantage in those cases.

The paper mentions the decision-theoretic and the empirical-utility framework, but then seems to consider only loss functions that are defined on the level of a single instance. Aren't the two settings that same in that case?

l. 525: Is the argmin unique? Are we breaking ties arbitrarily?

Despite being part of the theorem, $\mathcal{M}$ does not appear anywhere in the proof of 3.1

I tried going through the proof of 4.1, but I'm not quite sure how to construct the hypothesis $h'$ with that realized $s^{\mu}$, not do I see why the minimum is achieved for $s_h = s_y$, unless $c_h = c_y$.

**Limitations:**

I'm not sure if the proposed surrogate actually is tractable for label spaces with more than 50 labels.

---

> ### Author Rebuttal · Authors · 2024-08-06
>
> Thank you for your thoughtful feedback and suggestions on improving the readability. We will take them all into account when preparing the final version. Below, please find our responses to specific questions.
>
> **Weaknesses:**
>
> **1. The paper does not ...**
>
> **Response:** Thank you for your insightful comments.
>
> As you noted, our primary focus is theoretical and algorithmic: the design of
> theoretically principled surrogate losses for multi-label learning, supported by H-consistency bounds. We observe that some key previous publications on the topic, such as (Gao and Zhou, 2011), do not include any empirical result either. Nevertheless, we plan to include empirical results with commonly used multi-label surrogate losses in the final version of our paper, as you suggested.
>
> Our proposed framework offers significant progress by providing a unified surrogate loss that guarantees consistency for any multi-label loss. This contrasts with existing approaches, which achieve Bayes-consistency only for specific loss functions. While we have demonstrated the existence of efficient algorithms for minimizing multi-label logistic loss, we acknowledge the need for further exploration. Our future work will focus on extensive empirical analysis and the development of more universally applicable algorithmic solutions to cover a broader range of surrogate loss functions and diverse target losses.
>
> We agree with your observation that optimization is a crucial factor influencing the choice of surrogate losses and their practical performance, in addition to stronger consistency guarantees. The selection of surrogate losses and algorithms in practice should indeed consider multiple factors, including consistency, approximation, optimization properties, label dependency, and label correlation. We hope our work provides useful theoretical insights and potential alternatives helpful for this selection, with more extensive empirical analysis to follow.
>
> In conclusion, we believe our unified surrogate loss framework, which establishes strong consistency results for multi-label losses, represents a significant theoretical contribution. We are committed to further exploring the empirical aspects of this framework and developing practical solutions in future work.
>
> **2. It is claimed that ...**
>
> **Limitations: ...**
>
> **Response:** For many standard loss functions, the terms involving sums over the entire label space can be computed analytically. We illustrate this below for the Hamming loss and $F_{\beta}$ measure loss functions :$\sum_{y} ( 1 - \overline{ \mathsf L}_ {\mathrm{ham}}(y, y^j)  )= -(l - 1) 2^{l} + \sum_{y}  \sum_{i = 1}^{l} 1_{y_i = y^j_i} = 2^{l - 1} (2 - l)$, and $ \sum_{y} ( 1 - \overline{ \mathsf L}_ {F_ {\beta}}(y, y^j)  ) =  \sum_{y} \frac{(1 + \beta^2) y \cdot y^j }{\beta^2 \| y \|_1  +  \| y^j \|_1}$
>
> $= \sum_{k = 0}^{l}  \frac{(1 + \beta^2)  \| y^j \|_1\binom{l - 1}{k - 1} }{\beta^2 k + \| y^j \|_1}$. A similar analysis can be used for many other loss functions. Therefore, the presence of these terms does not impact the tractability of our algorithms. Additionally, as noted in Section 7, these terms can be precomputed once and reused, regardless of the specific sample or task under consideration.
>
> **Miscellaneous issues:**
>
> Thank you for pointing these out. We will number the equations, correct the typos and refine the statements accordingly.
>
> **Questions:**
>
> **1. In several ...**
>
> **Response:** In multi-label learning, label correlation simply means that certain pairs of labels (e.g., "cup" and "mug") tend to co-occur more frequently than others (e.g., "cup" and "umbrella") as labels for input points.  Leveraging these correlations can significantly enhance the efficiency of multi-label learning.
>
> Approaches like binary relevance surrogate loss treat each label independently, missing the opportunity to exploit these inherent relationships.  Our new form of surrogate losses directly takes into account such correlations among labels. Both the binary relevance surrogate loss and our new surrogate loss are Bayes-consistent, meaning that minimizing them over the family of all measurable functions approximates the Bayes-optimal solution. However, our new surrogate losses that consider label correlations can converge faster, which is reflected in their more favorable $H$-consistency bounds, independent of the number of labels.
>
> We will formalize these concepts and provide a more detailed discussion in the final version.
>
> **2. The paper mentions ...**
>
> **Response:** In the decision-theoretic analysis (DTA) framework, a loss function defined as a function over a single instance is considered, and the measure is defined as the expected loss, also known as the generalization error (expectation of a loss function over samples). In the empirical utility maximization (EUM) framework, the measures are directly defined as functions of the population (a function of an expectation over samples). In our paper, we adhere to the DTA framework by analyzing the loss functions and their consistency guarantees in multi-label learning.
>
> **3. l. 525 ...**
>
> **Response:** Any fixed deterministic strategy can be used to break ties. For example, we can choose the label with the lowest index under the natural ordering of labels as the tie-breaking strategy. We will elaborate on this in the final version.
>
> **4. Despite ...**
>
> **Response:** The minimizability gaps appear after taking the expectation on both sides of the inequality between lines 506 and 507, where we used the concavity of the function $\Gamma$ and Jensen's inequality. We will elaborate on this in the final version.
>
> **5. I tried ...**
>
> **Response:** The realization is due to the completeness assumption. The minimum is achieved for $\mathsf s_{\mathsf h}  = \mathsf s_{\mathsf y}$ because $\mathsf c_{\mathsf h} \geq  \mathsf c_{\mathsf y}$ and $\mathsf s_{\mathsf h} \geq  \mathsf s_{\mathsf y}$ by definition. We will elaborate on these in the final version.

---

> ### Comment · Reviewer_9NfQ · 2024-08-14
>
> To clarify, my main critique is not that the paper doesn't have any experiments; it is that it makes claims that extend beyond the purely theoretical, but these claims are not actually verified.
> To me, that could be resolved either way:
>  Remove these claims and have a pure theory paper
>  Add experiments that verify these claims
>
> For example, if you had written an article that introduced Strassen multiplication and claimed that it would lead to real-world speed-ups in matrix multiplication without providing an actual implementation, I would have found that problematic, it is _very_ difficult to implement Strassen on actual hardware so that it beats regular matrix multiplication algorithms.
>
> One of my concerns, I believe not addressed in the rebuttal, is that, as far as I can see, in the general case, the precomputation may in itself be exponential in the number of labels.
>
> Overall, though, I do think that most of these points can be addressed in a camera-ready version by more careful writing, and therefore I will raise my score.

---

### Decision · Program_Chairs · 2024-09-25

**Decision:**

Accept (poster)

**Comment:**

This paper studies H-consistency for multi-label learning. It has several main contributions: (i) establishing H-consistency for existing binary relevance with a surrogate loss, then strengthening existing results; (ii) it introduces a new loss called multi-label logistic loss and establishes its H-consistency; (ii) it makes extensions of H-consistency for multi-label logistic loss to a broad family of multi-label losses and extensions to a family of new surrogate loss called multi-label comp-sum losses.  The paper also makes efforts about how to compute the gradient of the proposed multi-label logistic loss. These contributions are considered as novel and interesting.  However, there are still some limitations raised about the paper: (i) lack of experiments, which is an issues mainly because it proposes a new loss and claims its computing is easy; (ii) it may not be practical to use the proposed multi-label logistic loss.  One natural loss that has been neglected in the paper is the extension of multi-class logistic loss, i.e., $-\sum_{i=1}^l y_i \log\frac{\exp(h_i(x))}{\sum_k\exp(h_k(x))}$. The author are encouraged to take the reviews into account in the revision, especially about the claim of efficient computation by either adding some experiments or weaken the claim.